# ENTROPY-PRESERVING REINFORCEMENT LEARNING

**Aleksei Petrenko**[*]
Apple

**Ben Lipkin**[*†]
MIT

**Kevin Chen**
Apple

**Erik Wijmans**
Apple

**Marco Cusumano-Towner**
Apple

**Raja Giryes**
Apple

**Philipp Krähenbühl**
Apple

## ABSTRACT

Policy gradient algorithms have driven many recent advancements in language model reasoning. An appealing property is their ability to learn from exploration on their own trajectories, a process crucial for fostering diverse and creative solutions. As we show in this paper, many policy gradient algorithms naturally reduce the entropy—and thus the diversity of explored trajectories—as part of training, yielding a policy increasingly limited in its ability to explore. In this paper, we argue that entropy should be actively monitored and controlled throughout training. We formally analyze the contributions of leading policy gradient objectives on entropy dynamics, identify empirical factors (such as numerical precision) that significantly impact entropy behavior, and propose explicit mechanisms for entropy control. These include REPO, a family of algorithms that modify the advantage function to regulate entropy, and ADAPO, an adaptive asymmetric clipping approach. Models trained with our entropy-preserving methods maintain diversity throughout training, yielding final policies that are more performant and retain their trainability for sequential learning in new environments.

## 1 INTRODUCTION

Online policy gradient reinforcement learning (RL) has become the standard for boosting the reasoning abilities of language models (Jaech et al., 2024; Comanici et al., 2025; Guo et al., 2025). This approach involves sampling trajectories from the current policy within a given environment and reward function, then using these to estimate a gradient that maximizes expected reward. Effective RL optimization requires balancing exploration and exploitation (Thrun, 1992; Sutton et al., 1998), where a robust learner should generate diverse trajectories to cover the spectrum of potential solutions. Maximum entropy reinforcement learning offers a framework for achieving this balance (Ziebart et al., 2008; Haarnoja et al., 2017; 2018; Eysenbach & Levine, 2022). While trivially the optimal solution to a finite Markov decision process (MDP) is a deterministic stationary policy, optimization over the intermediate landscape requires a balance of exploration and exploitation.

A common issue observed in online algorithms like GRPO (Shao et al., 2024) is entropy collapse. This phenomenon occurs when training excessively narrows the distribution around already high-probability solutions from the base model, neglecting other correct but less probable options. This often leads to premature convergence to a local optimum, enhancing `pass@1` relative to base model at the expense of `pass@k` (Shao et al., 2024; Dang et al., 2025; Yue et al., 2025). This challenge has spurred innovations in policy gradient algorithm design, e.g. directly optimizing for `pass@k` performance (Chen et al., 2025b). Concurrently, research has highlighted GRPO's training instability and the complex interplay between off-policy drift, importance weight clipping, and entropy, inspiring modifications such as DAPO (Yu et al., 2025) and GSPO (Zheng et al., 2025).

In this work, we argue that **entropy should be actively monitored and controlled throughout RL training**. We analyze entropy preservation as a unifying lens for understanding the successes of recent algorithms and propose explicit mechanisms for entropy control. An important observation from our work is that, while a correlation exists between final entropy and performance, a more

---

[*]co-first authorship.

[†]work performed during an internship at Apple.

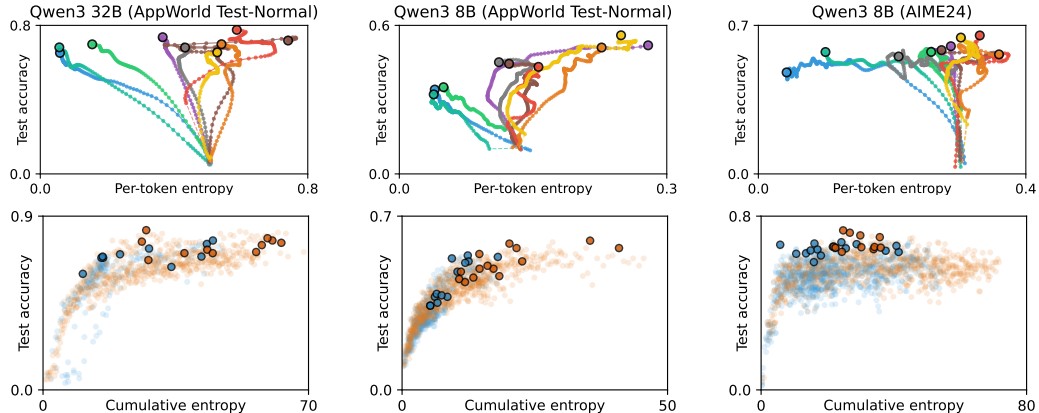

Figure 1: Top: Evolution of the average per-token entropy and test accuracy during training for several baselines (GRPO, LOOP, DAPO, GSPO) and their entropy regularized versions (REPO). Each curve shows the average trajectory over several training runs with different seeds. Bottom: Cumulative entropy experienced during training up to a given checkpoint is positively correlated with the test accuracy. Each point is a checkpoint of a single training run (best-performing checkpoint per run highlighted). Algorithms that collapse the entropy early (see e.g. `Qwen-3-8B` on AppWorld; middle column) perform significantly worse than algorithms that maintain a steady entropy during training.

informative measure is the entropy trajectory throughout the optimization process. As the saying goes, "it's not the destination, it's the journey." Figure 1 tracks this effect. A trajectory characterized by lower entropy throughout training yields lower performance. Conversely, if entropy trajectories are similar for most of the optimization but differ only in the final steps, performance is largely unaffected.

Our contributions span theory and algorithmic development. We analyze how policy gradient objectives modulate entropy dynamics, proving that PPO's clipping bounds entropy change and that DAPO's and GSPO's clipping implicitly preserve entropy. We identify critical implementation factors affecting entropy dynamics, including numerical precision (BF16 vs FP16) and framework behaviors (FSDP2 output casting), explaining previously observed training instabilities. We propose explicit entropy control mechanisms—REPO, which modifies the advantage function, and ADAPO, an adaptive asymmetric clipping approach—both using adaptive controllers to maintain target entropy levels. Our numerical fixes alone yield state-of-the-art on AppWorld (79% Test Normal, 71% Test Challenge), while entropy-preserving REPO and ADAPO achieve the strongest off-policy performance, closing the gap to on-policy training and retaining trainability for sequential learning.

## 2 PRELIMINARIES

**Language modeling.** Let $x \in \mathcal{X}$ denote the tokens in a vocabulary and $\boldsymbol{x} \in \mathcal{X}^*$ the strings expressible via concatenation of those tokens. A **language model** (LM) $\pi_\theta$ parameterized by $\theta$ defines a probability distribution over strings that factors autoregressively such that $\pi_\theta(\boldsymbol{x}) = \pi_\theta(\square \mid \boldsymbol{x}) \prod_{i=1}^{|\boldsymbol{x}|} \pi_\theta(x_i \mid \boldsymbol{x}_{<i})$, where $\square$ denotes an end of sequence (EOS) marker. For notational convenience we will use $\pi_\theta$ to express probabilities on both tokens and strings.

**Language modeling as a Markov decision process.** Let the policy $\pi_\theta$ sample **actions** $a \in \mathcal{A} = \mathcal{X} \cup \{\square\}$ (any token or EOS) given a **state** $\boldsymbol{s} \in \mathcal{X}^*$ (a string context). Let state transitions append generated actions to the state.[1] Let $\boldsymbol{\tau}$ denote a **trajectory**, a sequence of states and actions generated by the policy and environment. Let $\boldsymbol{\tau} \sim \pi_\theta$ denote the trajectory distribution. We consider tasks

---

[1]State transitions deterministically append the generated action to the context, terminating generation at EOS or upon some other environment condition. In some domains, e.g., those involving tool calls, state transitions may also append additional tokens to the state that were generated by some unobservable process such as executing a code interpreter.

with **terminal rewards** $R(\boldsymbol{c}, \boldsymbol{\tau})$. Given some task context $\boldsymbol{c}$ sampled from some dataset $\mathcal{D}$, the MDP objective is to maximize $\mathcal{J}_{\text{MDP}} \stackrel{\text{def}}{=} \mathbb{E}_{\boldsymbol{c} \sim \mathcal{D}, \boldsymbol{\tau} \sim \pi_\theta(\cdot | \boldsymbol{c})}[R(\boldsymbol{c}, \boldsymbol{\tau})]$.

**Policy gradient reinforcement learning** directly computes a gradient through the REINFORCE algorithm (Williams, 1992), which is amenable to Monte Carlo estimation:

$$\nabla_\theta \mathcal{J}_{\text{MDP}} = \mathbb{E}_{\boldsymbol{c} \sim \mathcal{D}, \boldsymbol{\tau} \sim \pi_\theta(\cdot | \boldsymbol{c})} \left[ A(\boldsymbol{c}, \boldsymbol{\tau}) \cdot \nabla_\theta \log \pi_\theta(\boldsymbol{\tau} \mid \boldsymbol{c}) \right],$$

where $A(\boldsymbol{c}, \boldsymbol{\tau}) = R(\boldsymbol{c}, \boldsymbol{\tau}) - b$ is an advantage function shifting the return $R(\boldsymbol{c}, \boldsymbol{\tau})$ by a baseline $b$.

**REINFORCE leave-one-out (RLOO)** (Kool et al., 2019; Ahmadian et al., 2024; Kazemnejad et al., 2024; Chen et al., 2025a) is one of the most popular estimates of advantage for language modeling. It generates $K$ independent samples on-policy $\boldsymbol{\tau}_1, \dots, \boldsymbol{\tau}_K \sim \pi_\theta(\cdot \mid \boldsymbol{c})$ for each task $\boldsymbol{c}$. The reward for each trajectory may then be baselined against the remaining $K - 1$ independent samples, yielding an unbiased, low variance advantage estimator:

$$\widehat{A}_{\text{RLOO}}(\boldsymbol{c}, \boldsymbol{\tau}_i) \stackrel{\text{def}}{=} R(\boldsymbol{c}, \boldsymbol{\tau}_i) - \frac{1}{K-1} \sum_{j=1}^{K} R(\boldsymbol{c}, \boldsymbol{\tau}_j) \mathbb{1}_{[i \neq j]} = \frac{K}{K-1} \left( R(\boldsymbol{c}, \boldsymbol{\tau}_i) - \frac{1}{K} \sum_{j=1}^{K} R(\boldsymbol{c}, \boldsymbol{\tau}_j) \right).$$

Policy gradient algorithms are on-policy by nature: They rely on a new set of trajectories in each context $\boldsymbol{\tau} \sim \pi_\theta(\cdot \mid \boldsymbol{c})$ after each gradient update of the policy $\pi_\theta$.

**Proximal policy optimization (PPO)** allows the updated policy to deviate slightly from a sampling policy (Schulman et al., 2017). It uses an importance weight to correct the magnitudes of parameter updates such that the expected policy gradient remains unbiased. These importance weights are typically clipped to avoid divergence from a local trust region (Schulman et al., 2015).

$$\mathcal{J}_{\text{PPO}} \stackrel{\text{def}}{=} \mathbb{E}_{\boldsymbol{c} \sim \mathcal{D}, \boldsymbol{\tau} \sim \pi_\theta(\cdot | \boldsymbol{c})} \left[ \frac{1}{|\boldsymbol{\tau}|} \sum_{a_t \in \boldsymbol{\tau}} \min \left( A(\boldsymbol{c}, \boldsymbol{\tau}) \cdot w_t, A(\boldsymbol{c}, \boldsymbol{\tau}) \cdot w_t |_{1-\epsilon}^{1+\epsilon} \right) \right] \quad w_t \stackrel{\text{def}}{=} \frac{\pi_\theta^{\text{new}}(a_t \mid \boldsymbol{c}, \boldsymbol{a}_{<t})}{\pi_\theta^{\text{old}}(a_t \mid \boldsymbol{c}, \boldsymbol{a}_{<t})},$$

where $w_t |_{1-\epsilon}^{1+\epsilon}$ clips the importance ratio from below $1 - \epsilon$ and above $1 + \epsilon$. In our theoretical analysis, we will examine PPO with and without clipping. The version studied will be clear from the context.

**LOOP** (Chen et al., 2025a) and **GRPO** (Shao et al., 2024) combine the above PPO objective with RLOO leave-one-out advantage estimates. GRPO rescales advantages by the standard deviation of the sample returns, introducing a small bias (Liu et al., 2025b), while LOOP uses $\widehat{A}_{\text{RLOO}}$ directly.

$$\widehat{A}_{\text{GRPO}}(\boldsymbol{c}, \boldsymbol{\tau}_i) \stackrel{\text{def}}{=} \frac{R(\boldsymbol{c}, \boldsymbol{\tau}_i) - \text{mean}(R(\boldsymbol{c}, \boldsymbol{\tau}_1), \dots, R(\boldsymbol{c}, \boldsymbol{\tau}_K))}{\text{std}(R(\boldsymbol{c}, \boldsymbol{\tau}_1), \dots, R(\boldsymbol{c}, \boldsymbol{\tau}_K))}$$

**Group Sequence Policy Optimization (GSPO)** Zheng et al. (2025) uses a trajectory-level trust region defined by the geometric average of a sequence's probability ratios

$$\mathcal{J}_{\text{GSPO}} \stackrel{\text{def}}{=} \mathbb{E}_{\boldsymbol{c} \sim \mathcal{D}, \boldsymbol{\tau} \sim \pi_\theta(\cdot | \boldsymbol{c})} \left[ \min \left( A(\boldsymbol{c}, \boldsymbol{\tau}) \cdot w^{\text{GSPO}}, A(\boldsymbol{c}, \boldsymbol{\tau}) \cdot w^{\text{GSPO}} |_{1-\epsilon}^{1+\epsilon} \right) \right] \quad w^{\text{GSPO}} \stackrel{\text{def}}{=} \left( \frac{\pi_\theta^{\text{new}}(\boldsymbol{\tau} \mid \boldsymbol{c})}{\pi_\theta^{\text{old}}(\boldsymbol{\tau} \mid \boldsymbol{c})} \right)^{\frac{1}{|\boldsymbol{\tau}|}}.$$

GSPO yields an equivalent gradient estimator to GRPO, LOOP, and RLOO on-policy, but clips tokens and trajectories differently as the updated policy $\pi_\theta^{\text{new}}$ drifts from the sampling policy $\pi_\theta^{\text{old}}$.

**Policy entropy.** The inherent uncertainty that a policy places over its generations may be expressed from an information theoretic standpoint as **entropy** – expected surprise: $\mathcal{H}_{\pi_\theta}(\mathcal{D}) = -\mathbb{E}_{\boldsymbol{c} \sim \mathcal{D}} \left[ \mathbb{E}_{\boldsymbol{\tau} \sim \pi_\theta(\cdot | \boldsymbol{c})} \left[ \log \pi_\theta(\boldsymbol{\tau} \mid \boldsymbol{c}) \right] \right]$. In addition to global entropy, we may consider the entropy over actions at any given state $\boldsymbol{s} = (\boldsymbol{c}, \boldsymbol{a}_{<t})$ as $\mathcal{H}_{\pi_\theta}(\cdot \mid \boldsymbol{s}) = -\mathbb{E}_{a \sim \pi_\theta(\cdot | \boldsymbol{s})} \left[ \log \pi_\theta(a \mid \boldsymbol{s}) \right]$.

In this paper, we analyze how state-wise entropy evolves as variants of policy gradient optimize their objectives. We identify which algorithm variants are naturally entropy preserving, and which lead to rapid collapse (§3). We demonstrate that subtle implementation details can distort entropy dynamics, causing unexpected collapse in algorithms that should theoretically preserve entropy (§4). Finally, we propose simple modifications of RL methods that lead to effective entropy regularization and improve downstream task performance (§5).

## 3 THEORY: ENTROPY DYNAMICS OF POLICY GRADIENT

The entropy dynamics of policy gradient RL boils down to the relationship between two values: (1) action log-probabilities, and (2) the advantages yielded by those actions. Intuitively, assigning a positive advantage to some action increases its probability. For high probability actions, this effect sharpens the distribution, and entropy decreases. For low probability actions, this flattens the distribution, increasing entropy. The opposite pattern holds for negative advantages. This effect is natural: after all, sharpening an uncertain policy around correct actions directly maximizes the expected return. However, as we will see, not all RL algorithms sharpen the distribution equally.

Formally, consider the policy gradient update with on-policy actions in state $s$. Under a first-order Taylor approximation to the training dynamics, the expected change in entropy is as follows.

**Theorem 1.** *Given a policy gradient update $\widehat{\theta} := \theta + \alpha \cdot \nabla_\theta \mathcal{J}_{\mathrm{MDP}}(s)$, the expected change in entropy is approximately:*

$$\Delta \mathcal{H}_{\pi_\theta}(\cdot \mid s) \approx -\alpha \cdot \mathbb{E}_{a \sim \pi_\theta(\cdot \mid s), a' \sim \pi_\theta(\cdot \mid s)} \left[ A(s, a) \cdot L(s, a') \cdot u(s, a)^\top u(s, a') \right].$$

$L(s, a) \stackrel{\text{def}}{=} \log \pi_\theta(a \mid s) - \mathbb{E}_{a \sim \pi_\theta(\cdot \mid s)}[\log \pi_\theta(a \mid s)]$ *denotes mean-centered log-probabilities and* $u(s, a) \stackrel{\text{def}}{=} \nabla_\theta \log \pi_\theta(a \mid s)$ *is the score function for a policy $\pi_\theta$ evaluated at state $s$ and action $a$.*

*[Proof in App. A.2].* The entropy change is driven by a multiplicative relationship between action log-probabilities and the advantages yielded by those actions. In an exact derivation, these are weighted by the score vector outer product. With additional independence assumptions or a tabular softmax policy parameterization, this expression can be further simplified, resulting in a weighting by the action probabilities. This yields the following corollary:

**Corollary 1.** *Assuming $u(s, a)^\top u(s, a') = 0$ for all $a \neq a'$, the entropy change is proportional to:*

$$\Delta \mathcal{H}_{\pi_\theta}(\cdot \mid s) \propto -\mathbb{E}_{a \sim \pi_\theta(\cdot \mid s)} \left[ A(s, a) \cdot L(s, a) \cdot \pi_\theta(a \mid s) \right]$$

*[Proof in App. A.3].* This latter form encodes the dominant behavior of entropy dynamics in a manner that is inherent to policy gradient. Using this form, we explain the observed behaviors of various RL algorithms. A similar derivation can be shown for tabular softmax policies (Cui et al., 2025, see Corollary 2 in App. A.4). Thm. 1 and Corollary 1 tell us that the change in entropy is governed by a correlation between advantages and log-probabilities, weighted by action probability.

**Entropy dynamics of PPO.** The biggest feature of PPO is its ability to train on slightly off-policy trajectories, given that the updated policy does not deviate from a trust region around the current policy. This allows PPO to take multiple policy-improvement steps for a single set of trajectories. The effect of these repeated updates is much larger policy updates between consecutive PPO steps, which empirically amplifies entropy collapse. This being said, the clipping on PPO, when appropriately orchestrated, can protect against entropy collapse as well. Clipping ensures that no policy gradient update is performed if the policy drifts outside a trust region $(1 - \epsilon_{\text{low}}) \cdot \pi_\theta^{\text{old}}(a \mid s) \leq \pi_\theta^{\text{new}}(a \mid s) \leq (1 + \epsilon_{\text{high}}) \cdot \pi_\theta^{\text{old}}(a \mid s)$. This bounds the change in entropy:

**Theorem 2.** *Proximal Policy Optimization (PPO) bounds the entropy $\mathcal{H}_{\pi_\theta^{new}}(\cdot \mid s)$ of the updated policy by the original policy entropy $\mathcal{H}_{\pi_\theta^{old}}(\cdot \mid s)$ such that:*

$$(1 - \epsilon_{low}) \cdot \mathcal{H}_{\pi_\theta^{old}}(\cdot \mid s) \leq \mathcal{H}_{\pi_\theta^{new}}(\cdot \mid s) \leq (1 + \epsilon_{high}) \cdot \mathcal{H}_{\pi_\theta^{old}}(\cdot \mid s)$$

*[Proof in App. A.5].* The clipping thresholds directly limit the maximum induced change in entropy per token. Intuitively, the change in entropy per token is stochastic: some actions have a large correlation between advantage and log probability; others do not, or even have an anti-correlation. For a symmetric clipping regime, this results in an entropy change that largely follows the statistical trends outlined above, but at a lower magnitude.

**Entropy dynamics of DAPO.** Now consider DAPO (Yu et al., 2025), with an asymmetric clipping regime $\epsilon_{\text{low}} < \epsilon_{\text{high}}$. This allows for larger entropy increases, while limiting the entropy decrease. Due to the stochastic nature of the entropy changes, this directly contributes to an overall increase in per-token entropy over sufficient samples. Threshold values $\epsilon_{\text{low}} = 0.2$ and $\epsilon_{\text{high}} = 0.28$ proposed in Yu et al. (2025) stabilize the entropy throughout training, as we show experimentally.

**Entropy dynamics of GSPO.** GSPO defines a trust region $1 - \epsilon_{\text{low}}^{\text{GSPO}} \leq w^{\text{GSPO}} \leq 1 + \epsilon_{\text{high}}^{\text{GSPO}}$, or equivalently $\left(1 - \epsilon_{\text{low}}^{\text{GSPO}}\right)^{|\boldsymbol{\tau}|} \leq \frac{\pi_\theta^{\text{new}}(\boldsymbol{\tau}|\boldsymbol{c})}{\pi_\theta^{\text{old}}(\boldsymbol{\tau}|\boldsymbol{c})} \leq \left(1 + \epsilon_{\text{high}}^{\text{GSPO}}\right)^{|\boldsymbol{\tau}|}$. This induces an equivalent bound to Thm. 2; however, the bound now depends on the trajectory length $|\boldsymbol{\tau}|$. Longer trajectories may induce a larger change in entropy, shorter trajectories induce a smaller change in entropy. With parameter values suggested in Zheng et al. (2025), $\epsilon_{\text{low}}^{\text{GSPO}} = 3 \times 10^{-4}$ and $\epsilon_{\text{high}}^{\text{GSPO}} = 4 \times 10^{-4}$, the entropy bound is tighter for trajectories $|\boldsymbol{\tau}| < \frac{\ln(1\pm\epsilon)}{\ln(1\pm\epsilon^{\text{GSPO}})} \approx 600$ tokens compared to DAPO. Like DAPO, the clipping range is asymmetric $\epsilon_{\text{low}}^{\text{GSPO}} < \epsilon_{\text{high}}^{\text{GSPO}}$ leading to a stochastic increase in entropy.

**Summary.** The theoretical analysis above reveals that entropy dynamics in policy gradient algorithms are governed by the correlation between advantages and log-probabilities. PPO's multiple off-policy updates amplify entropy collapse, while clipping mechanisms can bound the entropy change per update. Asymmetric clipping (DAPO) and sequence-level clipping (GSPO) provide implicit entropy preservation by allowing larger entropy increases than decreases. However, these implicit mechanisms may not be sufficient in all settings.

Importantly, even strictly on-policy algorithms like RLOO are subject to the entropy dynamics described in Corollary 1: if the base policy is already well-calibrated to the reward function, the correlation between advantages and log-probabilities will be positive, and entropy will decrease. RLOO avoids the *amplification* of this effect that arises from off-policy drift and repeated updates on recycled advantages, but does not eliminate the underlying dynamic. This explains why RLOO retains more entropy than PPO-based algorithms in most settings, yet can still exhibit meaningful entropy loss when the base model is strongly pre-calibrated to the task. Explicit entropy control mechanisms, which we present in §5, can therefore be valuable even in on-policy settings.

## 4 EMPIRICAL FINDINGS: IMPLEMENTATION DETAILS AFFECTING ENTROPY

We identify empirical factors that significantly impact entropy dynamics, discussed in this section.

### 4.1 16-BIT QUANTIZATION OF MODEL OUTPUTS AFFECTS CLIPPING

In PPO, GRPO, DAPO and similar methods, the clipped objective requires computing the probability ratio: $r = \frac{\pi_\theta(a|s)}{\pi_{\theta_{\text{old}}}(a|s)}$. By default, common LLM training stacks (e.g. *HF Accelerate & FSDP2*; App. B.1) cast model outputs to the training dtype before downstream computations. As a result, the probability ratio is ultimately computed as: $r_{\text{observed}} = \exp(\text{bf16}(\log \pi_\theta(a|s)) - \text{bf16}(\log \pi_{\theta_{\text{old}}}(a|s)))$ where $\text{bf16}(\cdot)$ denotes bfloat16 casting with round-to-nearest-even.

**Theorem 3.** *Under bf16 quantization, the observed ratio exhibits a multiplicative upward bias: $\mathbb{E}[r_{\text{observed}} \mid r_{\text{true}}] > r_{\text{true}}$. [Proof in App. A.8].*

Note that the bias is **proportional to $r_{\text{true}}$**: larger ratios experience proportionally larger absolute bias. This creates an effective asymmetric clipping that systematically favors entropy decrease:

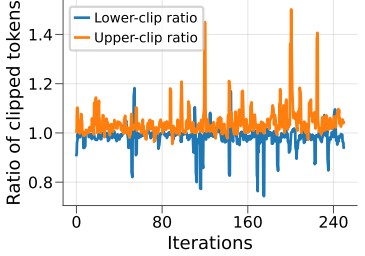

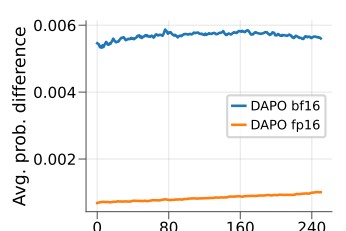

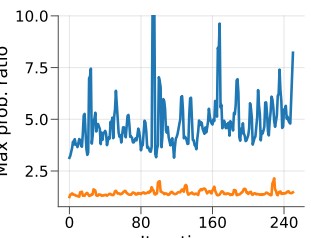

(a) 16-bit $\log \pi$ quantization effects.    (b) Inference/train discrepancy depending on dtype.

Figure 2: (a) Fraction of tokens hitting clip bounds with 16-bit rounding: $\epsilon_{\text{high}}$ is reached more often and $\epsilon_{\text{low}}$ less often, hindering the promotion of low-probability actions. (b) Average probability difference and max importance weight ratio between vLLM inference and training forward pass.

**Upper clip**: For advantageous actions ($A > 0$, $r > 1$), the upward bias causes $r_{\text{observed}}$ to reach the upper clip earlier, limiting probability increases.

**Lower clip**: For disadvantageous actions ($A < 0$, $r < 1$), the upward bias pulls $r_{\text{observed}}$ toward 1, making the lower clip less restrictive. Fig. 2a empirically shows how $\epsilon_{\text{high}}$ is reached more frequently and $\epsilon_{\text{low}}$ less frequently with quantization compared to the full-precision calculation.

This behavior is equivalent to asymmetric clipping with $\epsilon_{\text{low}} > \epsilon_{\text{high}}$, the opposite direction from DAPO's entropy-preserving asymmetry! While this bias exists in any finite-precision implementation, very limited precision of BF16 promotes this to a strong entropy-decreasing effect (App. B.1). A trivial solution for this problem is to simply use full precision for model outputs throughout the computation graph, including any importance ratio calculations.

## 4.2 FLOAT16 VS BFLOAT16 TRAINING

It is customary to use the BF16 floating-point type in LLM training because of its larger dynamic range. However, Qi et al. (2025) report improved results with *float16* (FP16) as its additional mantissa bits enable more accurate gradient representation. Using FP16 format significantly reduces the discrepancies between the LLM inference (vLLM) and training subsystem, inherent to the modern post-training stacks (Fig. 2b).

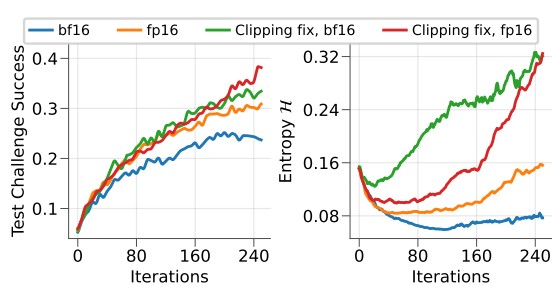

Figure 3: FP16 training on `Qwen-3-8B` AppWorld and clipping fix lead to a qualitative change: DAPO entropy collapse transitions to entropy increase.

In practice, with appropriate loss and gradient scaling, FP16 training tends to mitigate entropy collapse and yield more stable and predictable training. To highlight the importance of these empirical findings: FP16 training in conjunction with the $\log \pi_\theta$ rounding fix (§4.1) results in qualitatively different entropy dynamics, allowing DAPO's entropy-increasing asymmetric clipping to overcome collapse (Fig. 3).

## 5 EXPLICIT ENTROPY CONTROL METHODS

The theory in §3 and empirical analysis in §4 reveal that entropy dynamics are influenced by many factors and that minor implementation details can qualitatively change the algorithm behavior. While implicit mechanisms (asymmetric or sequence-level clipping) provide some level of control, an explicit entropy regulation technique may be required for stable RL post-training.

A standard approach to entropy preservation is adding an explicit entropy bonus to the objective: $\mathcal{J}_{\text{entropy}} = \mathcal{J}_{\text{PPO}} + \beta \cdot \mathcal{H}_{\pi_\theta}$. However, this approach has significant drawbacks:

**Fixed coefficient**: A fixed $\beta$ does not account for the evolving entropy dynamics over training.

**Memory cost**: Computing the entropy bonus exactly requires materializing all logits, which is memory-intensive for large vocabularies, especially compared to memory-efficient methods such as Cut Cross-Entropy (CCE, Wijmans et al., 2025).

Below, we address both issues by proposing an adaptive entropy controller and a paired-sampling estimator that jointly estimates the policy and entropy gradients without materializing full logits.

### 5.1 REPO: REGULATED ENTROPY POLICY OPTIMIZATION

We propose REPO (Regulated Entropy Policy Optimization), which modifies the advantage function to include a scaled policy log-likelihood term: $A_{\text{REPO}}(\boldsymbol{s}, a) = A(\boldsymbol{s}, a) - \beta_{\boldsymbol{s}} \cdot L(\boldsymbol{s}, a)$ for each $\boldsymbol{s} = (\boldsymbol{c}, \boldsymbol{a}_{<t})$. This updated advantage is no longer constant throughout the trajectory like in RLOO and variants, but differs for individual tokens $a_t \in \boldsymbol{\tau}$.

Following Thm. 1 by Prop. 3, the induced change in entropy with $A_{\text{REPO}}$ is:

$$\Delta \mathcal{H}_{\pi_\theta}^{\text{REPO}}(\cdot \mid \boldsymbol{s}) \approx \Delta \mathcal{H}_{\pi_\theta}(\cdot \mid \boldsymbol{s}) + \beta_{\boldsymbol{s}} \cdot \underbrace{\alpha \cdot \left\| \mathbb{E}_{a \sim \pi_\theta(\cdot \mid \boldsymbol{s})} \left[ L(\boldsymbol{s}, a) \cdot u(\boldsymbol{s}, a) \right] \right\|^2}_{\geq 0}.$$

This provides a direct mechanism to control entropy where $\beta_{\boldsymbol{s}} > 0$ increases entropy relative to the default dynamic and $\beta_{\boldsymbol{s}} < 0$ downregulates entropy.

**REPO-D (Decorrelate).** One natural choice is to counteract entropy collapse on a per-token level by setting $\beta_{\boldsymbol{s}}^{\text{REPO-D}} \propto -\Delta\mathcal{H}_{\pi_\theta}(\cdot \mid \boldsymbol{s})$ as approximated in Corollary 1. This neutralizes $\Delta\mathcal{H}_{\pi_\theta}$, allowing $\Delta\mathcal{H}_{\pi_\theta}^{\text{REPO}}$ to approach 0.

**REPO-R (Rescale).** REPO-R is an efficient practical approximation that captures the core intuition: increasing policy entropy requires upweighting rare correct solutions, while reducing (on average) the penalty assigned to rare incorrect solutions. This is accomplished by rescaling advantages based on action probabilities via $\beta_{\boldsymbol{s},a}^{\text{REPO-R}} = \zeta \, |A(\boldsymbol{s}, a)|$. The derivation from the general REPO advantage and implementation details are provided in App. D.2.

**Adaptive controller.** The optimal scale of the regularizer depends on learning rate, gradient structure, second-order effects, etc. We use a simple adaptive bidirectional controller:

1. Estimate $\mathcal{H}_{\pi_\theta}^{\text{init}}$, the policy entropy over experience collected in the first iteration; set $\zeta \leftarrow 10^{-3}$.

2. Each iteration: if $\mathcal{H}_{\pi_\theta} < \mathcal{H}_{\pi_\theta}^{\text{init}}$, update $\zeta \leftarrow \zeta \times 2$; if $\mathcal{H}_{\pi_\theta} > \mathcal{H}_{\pi_\theta}^{\text{init}}$, update $\zeta \leftarrow \zeta \div 2$.

3. Flip the sign of $\zeta$ to exert pressure in the opposite direction: if $|\zeta| < \zeta_{\min}$ set $\zeta \leftarrow -\zeta$.

4. Clip magnitude of $|\zeta|$ to $[\zeta_{\min}, \zeta_{\max}]$. In our experiments we used $[10^{-3}, 10]$ for REPO-D and $[10^{-4}, 0.05]$ for REPO-R.

**Efficient estimation.** Both REPO-D and REPO-R can be effectively estimated using only the log-probability of the *sampled* token $a_t$, which is already available during the forward pass when using CCE (Wijmans et al., 2025). This stands in contrast to an explicit entropy bonus, which requires materializing the full logit vector over the vocabulary. We show in App. A.7 that REPO-D is formally equivalent to such an entropy bonus, but estimated via REINFORCE using paired samples, incurring **zero additional memory cost** and acting as a **control variate** that reduces gradient variance when advantages and log-probabilities are positively correlated, which is typical.

## 5.2 ADAPO: ADAPTIVE ASYMMETRIC CLIPPING

An alternative approach is to dynamically adjust the asymmetric clipping thresholds, taking advantage of DAPO's entropy-preserving properties. ADAPO (Adaptive DAPO) keeps $\epsilon_{\text{low}} = 0.2$ fixed and initializes $\epsilon_{\text{high}} = 0.28$ (as in DAPO), then varies $\epsilon_{\text{high}}$ in the range $[0.2, 0.32]$ based on the observed entropy: (1) If $\mathcal{H}_{\pi_\theta} < \mathcal{H}_{\pi_\theta}^{\text{init}}$: $\epsilon_{\text{high}} \leftarrow \min(1.05 \times \epsilon_{\text{high}}, 0.32)$ (allow more entropy increase) (2) If $\mathcal{H}_{\pi_\theta} > \mathcal{H}_{\pi_\theta}^{\text{init}}$: decrease $\epsilon_{\text{high}} \leftarrow \max(0.95 \times \epsilon_{\text{high}}, 0.2)$ (limit entropy increase). This provides bidirectional control over entropy through the clipping mechanism. Note that REPO can be used with many popular RL methods (RLOO, GRPO, DAPO, etc.) while ADAPO utilizes asymmetric clipping and is relevant to methods like DAPO or GSPO. In our experiments, we evaluate REPO-R on top of GRPO and ADAPO on top of DAPO (§6). RL methods using adaptive asymmetric clipping were independently proposed in contemporaneous work (Xi et al. (2025)).

## 6 EXPERIMENTS

We evaluate whether entropy-preserving training yields improvements to strong models on challenging environments when compared to state-of-the-art learning algorithms. We choose `Qwen-3-8B` and `Qwen-3-32B` as our starting policies (Yang et al., 2025).

**Environments.** *Interactive tool-use agent.* Training scenarios are drawn from the train split (90 problems) of the AppWorld benchmark (Trivedi et al., 2024). The AppWorld Test Normal (*TN*, 168 tasks) and Test Challenge (*TC*, 417 tasks) splits are used for evaluation. Terminal reward is calculated via task-provided unit-tests that check the final state of the environment against ground truth (additional details in App. C.1). *Competition-level mathematics.* Training scenarios are drawn from a non-overlapping quality-filtered subset of the AMC/AIME section of NuminaMath-1.5 (563 problems; Li et al., 2024). AIME 2024 (30 problems) and AIME 2025 (30 problems) are used as

evaluation datasets. Terminal reward indicates whether the generated answer matches the reference. We note that recent models are significantly overfit to math benchmarks so we strictly limit token budget to 4096 in AIME to create a challenging learning problem.

**Algorithms.** For each algorithm, we highlight its distinguishing features with otherwise minimal deviations from the base policy gradient to aid reproducibility (thus, some details and hyperparameter choices may differ slightly from original sources).

*RLOO*: REINFORCE with the $\widehat{A}_{\text{RLOO}}$ advantage estimator. Training is strictly on-policy (1 epoch) with a large minibatch that comprises all collected experience. *GRPO*: Off-policy extension of RLOO that uses normalized Leave None Out (LNO) estimator $\widehat{A}_{\text{GRPO}}$ and symmetric PPO clipping with $\epsilon = 0.2$. With GRPO and all algorithms below we train on the collected experience for 2 epochs with the minibatch size of 128 (AppWorld) or 256 (AIME) trajectories. *LOOP*: A variant of GRPO with non-normalized estimator $\widehat{A}_{\text{RLOO}}$ (RL SOTA on AppWorld at the time of writing). *DAPO*: a variant of LOOP with asymmetric clipping ($\epsilon_{\text{low}} = 0.2$, $\epsilon_{\text{high}} = 0.28$). *GSPO*: An adjustment of DAPO using $w^{\text{GSPO}}$ importance weighting and trajectory-based clipping with $\epsilon_{\text{low}}^{\text{GSPO}} = 3 \times 10^{-4}$ and $\epsilon_{\text{high}}^{\text{GSPO}} = 4 \times 10^{-4}$. *REPO-R*: modification of GRPO with the additional entropy control mechanism. *ADAPO*: Adaptive DAPO that adjusts $\epsilon_{\text{high}}$ based on observed entropy dynamics (§5.2).

## 6.1 VARIABLE ENTROPY DYNAMICS ACROSS ALGORITHMS

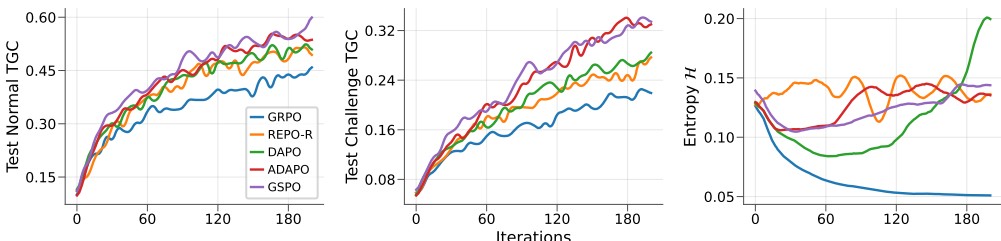

Figure 4: Entropy-preserving methods compared to baselines with `Qwen-3-8B` on AppWorld.

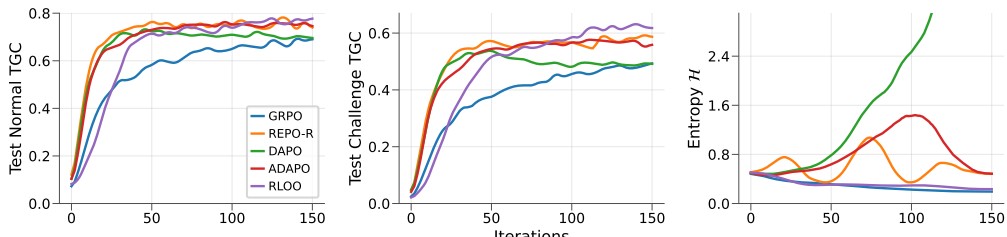

Figure 5: Entropy-preserving RL training with `Qwen-3-32B` on AppWorld.

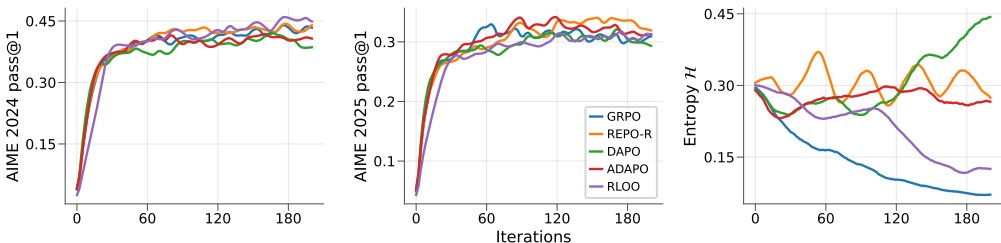

Figure 6: Entropy-preserving RL training with `Qwen-3-8B` on AIME.

We observe consistent patterns across AppWorld (Figs. 4 and 5) and AIME experiments (Fig. 6):

**PPO-like algorithms deplete entropy faster than strictly on-policy.** GRPO reduces entropy by nearly 90% over training, while RLOO loses considerably less. LOOP behaves very similarly to GRPO and thus omitted for readability. See comprehensive results summary in App. C.3.

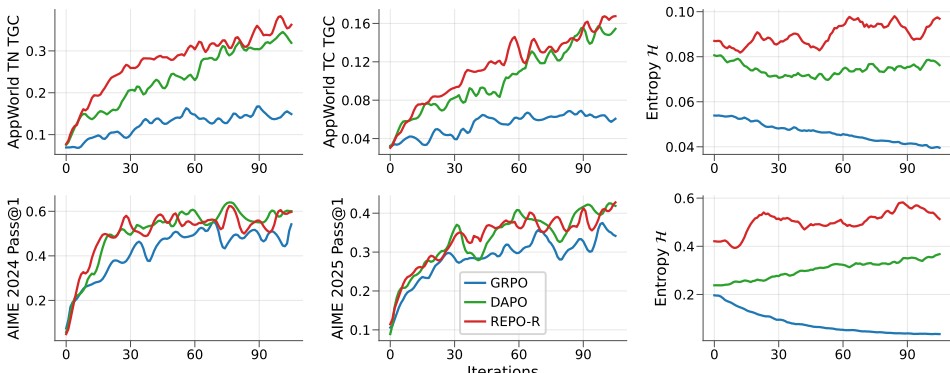

Figure 7: Sequential learning experiment. Top row: We use an AIME-trained model for GRPO, DAPO, REPO-R, and continue training the model on AppWorld. The left and middle plots show Task Goal Completion (TGC) on the normal (TN) and challenging (TC) test sets. A collapsed model (GRPO) does significantly worse than one in which entropy is preserved. Bottom row: We use an AppWorld-trained model and continue training on AIME. The same trends hold. All curves reflect the mean across three independent seeds.

**Clipping modifications protect entropy.** Following the intuition provided in §3, DAPO and GSPO retain considerably more entropy. Confirming our observations in §4, DAPO's entropy can uncontrollably increase in some experiments without an entropy-control mechanism (Fig. 5).

**Entropy-preserving methods outperform baselines.** REPO-R and ADAPO score higher than their off-policy baselines (GRPO and DAPO) and maintain steady policy entropy throughout training.

## 6.2 ENTROPY PRESERVATION AND DOWNSTREAM PERFORMANCE

We evaluate the effect of entropy preservation on downstream performance. See Fig. 1 for a preview of these results. We find that methods that preserve per-token entropy, maintaining a higher cumulative entropy over training, yield higher final test accuracy than those that don't. These trends are stronger on AppWorld than AIME. We hypothesize that `Qwen-3` family of models is heavily optimized for AIME, and so this optimization may have primarily involved sharpening around existing solutions. AppWorld, on the other hand, requires considerable exploration to discover new capabilities.

## 6.3 ENTROPY PRESERVATION ASSISTS SEQUENTIAL TRAINING

We evaluate how well different algorithms support further RL fine-tuning on a different task (i.e. sequential training). To that end, we first train `Qwen-3-8B` on either the AIME or AppWorld. We then take the best checkpoint as the starting point for training on the opposing environment. Fig. 7 shows that policies trained with GRPO perform poorly in the second training stage: due to entropy collapse, they lose their ability to explore. On the other hand, DAPO, and especially REPO, start re-training with ample entropy and retain their exploration ability over the course of training.

## 6.4 NUMERICAL PRECISION STABILIZES ENTROPY AND PERFORMANCE

Figure 3 shows that for `Qwen-3-8B` AppWorld training the numerical fixes have a dramatic impact: DAPO, which previously exhibited entropy collapse in this setting, now shows rapid entropy increase as the analysis of its asymmetric clipping design suggests. This shows that the observed entropy dynamics are highly sensitive to implementation details that may not be immediately apparent, and that some previously reported entropy collapse phenomena may have been artifacts of numerical precision rather than fundamental properties of algorithms.

**RLOO achieves state-of-the-art performance.** After switching to FP16 training (§4), purely on-policy RLOO sets the highest score at the time of submission on the AppWorld benchmark: our best checkpoint scored **79% Test Normal** and **71% Test Challenge** with `Qwen-3-32B`.

## 7 RELATED WORK

Reinforcement learning has emerged as the dominant paradigm for aligning pre-trained language models (Ziegler et al., 2019; Stiennon et al., 2020; Ouyang et al., 2022). This approach has been successfully scaled in environments yielding verifiable rewards such as programming and mathematics (Jaech et al., 2024; Lambert et al., 2024; Comanici et al., 2025; Guo et al., 2025; Team et al., 2025).

Empirically, training in this setting has typically been viewed as sharpening the base policy around existing solutions rather than yielding new ones (Gandhi et al., 2025; Liu et al., 2025b; Yue et al., 2025; Zhao et al., 2025). A good pre-trained base policy starts off already calibrated to many reasonable reward functions, and post-training can be viewed as tempering this distribution (Kadavath et al., 2022; Cui et al., 2025). In fact, several works directly exploit this calibration to drive accuracy improvements via unsupervised post-training. Agarwal et al. (2024) simply minimize entropy, Prasad et al. (2024); Zhang et al. (2025); Zuo et al. (2025) align to the model's majority vote distribution, Wang et al. (2025) get by with a single labeled sample, and Shao et al. (2025) even use random rewards. All of these works can be explained by simply allowing policy gradient to sharpen an already calibrated base policy. While this type of approach can help `pass@1`, it harms `pass@k` (Shao et al., 2024; Dang et al., 2025; Yue et al., 2025).

Some works protect against this pathological entropy collapse using modified policy gradient objectives. He et al. (2025) add auxiliary rewards to solutions as a function of their probability rank within a batch. Yu et al. (2025) introduce wider PPO clipping to encourage stronger reinforcement of low probability correct actions. Zheng et al. (2025) propose sequence-level clipping more independent of individual action probabilities. Chen et al. (2025b) reformulate online policy gradient to optimize `pass@k` as opposed to `pass@1`. Most similarly to our work, Cui et al. (2025); Xi et al. (2025); Wang et al. (2026) derive theoretical results regarding the covariance between advantages and probabilities mediating entropy collapse and then propose different approaches to counter this.

Other works impose a $D_{\text{KL}}$ penalty during training as an approach for preserving the base policy (e.g., Ziegler et al., 2019; Guo et al., 2025, etc.). However, it has been shown that such an approach limits how much the policy can learn (Korbak et al., 2022; Yang et al., 2024; Wu & Choi, 2025). For this reason, Chen et al. (2025a); Yu et al. (2025) remove the $D_{\text{KL}}$ penalty, (Vassoyan et al., 2025) ignore it for a subset of tokens, and (Liu et al., 2025a) iteratively reset the reference policy.

## 8 CONCLUSION

In this work, we argue that entropy should be actively monitored and controlled throughout reinforcement learning training for language models. We provide a theoretical analysis showing how policy gradient objectives modulate entropy dynamics, explaining why algorithms like GRPO exhibit entropy collapse while DAPO and GSPO provide implicit preservation. We identify critical empirical factors, notably numerical precision (BF16 vs FP16) and framework behaviors (FSDP2 output casting), that impact entropy dynamics and training instabilities. Building on these insights, we propose explicit mechanisms for entropy control: REPO, which modifies the advantage function, and ADAPO, which adaptively adjusts clipping thresholds. Our entropy-preserving methods perform strongly on AIME and AppWorld, outperforming their baseline counterparts (GRPO and DAPO) and improving sequential learning. We also report state-of-the-art results on AppWorld at the time of submission (79% Test Normal, 71% Test Challenge with RLOO and FP16 training).

We identify a distinction between **strictly on-policy** algorithms like RLOO and **weakly on-policy** algorithms like GRPO and GSPO. Our results show that, with proper numerical handling, strictly on-policy RLOO achieves the best performance overall. However, strictly on-policy training requires synchronous updates, which creates a bottleneck in distributed systems. Weakly on-policy methods enable asynchronous training pipelines where trajectory collection and policy updates can proceed in parallel, significantly improving throughput. The entropy-preservation mechanisms we propose (REPO, ADAPO) are compatible with both paradigms and can help weakly on-policy methods approach the performance of strictly on-policy training while maintaining the throughput benefits of asynchronous execution.

Overall, we highlight that entropy (and the corresponding exploration capability) is crucial for effective policy optimization and should be treated as a first-class concern in RL training pipelines.

ETHICS STATEMENT

This paper investigates the properties of policy gradient algorithms for language model reasoning, specifically focusing on the tendency for entropy collapse during training. Our research is primarily theoretical and analytical, involving mathematical analysis and algorithm development. Our work aims to improve entropy during reinforcement learning, which can lead to better exploration and wider diversity in generated outputs. We acknowledge the potential for misuse of advanced language models, including the generation of biased, harmful, or misleading content. We believe that responsible research practices, including transparency in model limitations and potential societal impacts, are crucial for mitigating these risks, and we hope that our research contributes to the development of more robust, creative, and beneficial language models.

REPRODUCIBILITY STATEMENT

Complete proofs for all theoretical claims, along with experimental details and hyperparameters, are included in the appendix. All data points presented in this work are the result of multiple repetitions of each experiment using independent random seeds.

USE OF LARGE LANGUAGE MODELS FOR WRITING

We acknowledge the use of large language models to assist with typographical corrections, phrasing, and self-review aimed at improving the clarity and structure of this manuscript.

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

## A PROOFS & DERIVATIONS

### A.1 BROADLY USED LEMMAS

**Lemma 1.** *The expected score function of policy $\pi_\theta$ at some state $\boldsymbol{s}$ is:*

$$\mathbb{E}_{a \sim \pi_\theta(\cdot|\boldsymbol{s})} \left[ \nabla_\theta \log \pi_\theta(a \mid \boldsymbol{s}) \right] = 0$$

*Proof.*

$$
\begin{aligned}
\mathbb{E}_{a \sim \pi_\theta(\cdot|\boldsymbol{s})} \left[ \nabla_\theta \log \pi_\theta(a \mid \boldsymbol{s}) \right] &= \sum_a \pi_\theta(a \mid \boldsymbol{s}) \cdot \nabla_\theta \log \pi_\theta(a \mid \boldsymbol{s}) \\
&= \sum_a \nabla_\theta \pi_\theta(a \mid \boldsymbol{s}) \\
&= \nabla_\theta \sum_a \pi_\theta(a \mid \boldsymbol{s}) \\
&= \nabla_\theta(1) \\
&= 0
\end{aligned}
$$

∎

**Lemma 2.** *The gradient of a sample estimate $\mathbb{E}_{x \sim P_\theta} \left[ f_\theta(x) \right]$ of function $f_\theta$ over distribution $P_\theta$ is:*

$$\nabla_\theta \mathbb{E}_{x \sim P_\theta} \left[ f_\theta(x) \right] = \mathbb{E}_{x \sim P_\theta} \left[ \nabla_\theta f_\theta(x) + f_\theta(x) \cdot \nabla_\theta \log P_\theta(x) \right]$$

*Proof.*

$$
\begin{aligned}
\nabla_\theta \mathbb{E}_{x \sim P_\theta} \left[ f_\theta(x) \right] &= \sum_x \nabla_\theta \left( P_\theta(x) \cdot f_\theta(x) \right) \\
&= \sum_x \left( P_\theta(x) \cdot \nabla_\theta f_\theta(x) + f_\theta(x) \cdot \underbrace{\nabla_\theta P_\theta(x)}_{P_\theta(x) \nabla_\theta \log P_\theta(x)} \right) \\
&= \sum_x P_\theta(x) \left( \nabla_\theta f_\theta(x) + f_\theta(x) \cdot \nabla_\theta \log P_\theta(x) \right) \\
&= \mathbb{E}_{x \sim P_\theta} \left[ \nabla_\theta f_\theta(x) + f_\theta(x) \cdot \nabla_\theta \log P_\theta(x) \right]
\end{aligned}
$$

∎

**Lemma 3.** *The gradient of a sample estimate $\mathbb{E}_{x \sim P_\theta} \left[ f_\theta(x) \right]$ of function $f_\theta$ over distribution $P_\theta$ can be baselined for any arbitrary $b$ independent of $x$:*

$$\nabla_\theta \mathbb{E}_{x \sim P_\theta} \left[ f_\theta(x) - b \right] = \nabla_\theta \mathbb{E}_{x \sim P_\theta} \left[ f_\theta(x) \right]$$

*Proof.*

$$
\begin{aligned}
\nabla_\theta \mathbb{E}_{x \sim P_\theta} \left[ f_\theta(x) - b \right] &= \mathbb{E}_{x \sim P_\theta} \left[ (f_\theta(x) - b) \cdot \nabla_\theta \log P_\theta(x) \right] \\
&= \mathbb{E}_{x \sim P_\theta} \left[ f_\theta(x) \cdot \nabla_\theta \log P_\theta(x) \right] - \mathbb{E}_{x \sim P_\theta} \left[ b \cdot \nabla_\theta \log P_\theta(x) \right] \\
&= \mathbb{E}_{x \sim P_\theta} \left[ f_\theta(x) \cdot \nabla_\theta \log P_\theta(x) \right] - b \cdot \underbrace{\mathbb{E}_{x \sim P_\theta} \left[ \nabla_\theta \log P_\theta(x) \right]}_{0} \\
&= \mathbb{E}_{x \sim P_\theta} \left[ f_\theta(x) \cdot \nabla_\theta \log P_\theta(x) \right] \\
&= \nabla_\theta \mathbb{E}_{x \sim P_\theta} \left[ f_\theta(x) \right]
\end{aligned}
$$

∎

**Lemma 4.** *The gradient of MDP objective $\mathcal{J}_{\mathrm{MDP}}$ at some state $\boldsymbol{s}$ is:*

$$\nabla_\theta \mathcal{J}_{\mathrm{MDP}}(\boldsymbol{s}) = \mathbb{E}_{a \sim \pi_\theta(\cdot|\boldsymbol{s})} \left[ (R(\boldsymbol{s}, a) - b) \cdot \nabla_\theta \log \pi_\theta(a \mid \boldsymbol{s}) \right]$$

*for any arbitrary baseline $b$ independent of $a$.*

*Proof.* Largely following (Williams, 1992), Lemma 2, and Lemma 3

$$
\begin{aligned}
\nabla_\theta \mathcal{J}_{\mathrm{MDP}}(\boldsymbol{s}) &= \nabla_\theta \mathbb{E}_{a\sim\pi_\theta(\cdot|\boldsymbol{s})}\left[R(\boldsymbol{s},a)\right] \\
&= \nabla_\theta \mathbb{E}_{a\sim\pi_\theta(\cdot|\boldsymbol{s})}\left[(R(\boldsymbol{s},a)-b)\right] \\
&= \mathbb{E}_{a\sim\pi_\theta(\cdot|\boldsymbol{s})}\left[(R(\boldsymbol{s},a)-b)\cdot\nabla_\theta\log\pi_\theta(a\mid\boldsymbol{s})\right] + \underbrace{\mathbb{E}_{a\sim\pi_\theta(\cdot|\boldsymbol{s})}\left[\nabla_\theta\left(R(\boldsymbol{s},a)-b\right)\right]}_{0} \\
&= \mathbb{E}_{a\sim\pi_\theta(\cdot|\boldsymbol{s})}\left[(R(\boldsymbol{s},a)-b)\cdot\nabla_\theta\log\pi_\theta(a\mid\boldsymbol{s})\right]
\end{aligned}
$$

∎

**Lemma 5.** *The gradient of the policy entropy at some state $\boldsymbol{s}$ is:*

$$
\nabla_\theta \mathcal{H}_{\pi_\theta}(\cdot\mid\boldsymbol{s}) = -\mathbb{E}_{a\sim\pi_\theta(\cdot|\boldsymbol{s})}\left[(\log\pi_\theta(a\mid\boldsymbol{s})-b)\cdot\nabla_\theta\log\pi_\theta(a\mid\boldsymbol{s})\right]
$$

*for any arbitrary baseline $b$ independent of $a$.*

*Proof.* Follows directly from Lemma 4 with $R(\boldsymbol{s},a) = -\log\pi_\theta(a\mid\boldsymbol{s})$.

$$
\begin{aligned}
\nabla_\theta \mathcal{H}_{\pi_\theta}(\cdot\mid\boldsymbol{s}) &= -\nabla_\theta\mathbb{E}_{a\sim\pi_\theta(\cdot|\boldsymbol{s})}\left[\log\pi_\theta(a\mid\boldsymbol{s})\right] \\
&= -\mathbb{E}_{a\sim\pi_\theta(\cdot|\boldsymbol{s})}\left[(\log\pi_\theta(a\mid\boldsymbol{s})-b)\cdot\nabla_\theta\log\pi_\theta(a\mid\boldsymbol{s})\right]
\end{aligned}
$$

∎

**Lemma 6.** *The expected advantage function $A(\boldsymbol{s},a) \overset{\mathrm{def}}{=} R(\boldsymbol{s},a) - b$, with baseline $V(\boldsymbol{s}) \overset{\mathrm{def}}{=} \mathbb{E}_{a\sim\pi_\theta(\cdot|\boldsymbol{s})}[R(\boldsymbol{s},a)]$, at some state $\boldsymbol{s}$ is:*

$$
\mathbb{E}_{a\sim\pi_\theta(\cdot|\boldsymbol{s})}[A(\boldsymbol{s},a)] = 0
$$

*Proof.*

$$
\begin{aligned}
\mathbb{E}_{a\sim\pi_\theta(\cdot|\boldsymbol{s})}[A(\boldsymbol{s},a)] &= \mathbb{E}_{a\sim\pi_\theta(\cdot|\boldsymbol{s})}[R(\boldsymbol{s},a)-V(\boldsymbol{s})] \\
&= \mathbb{E}_{a\sim\pi_\theta(\cdot|\boldsymbol{s})}[R(\boldsymbol{s},a)]-V(\boldsymbol{s}) \\
&= V(\boldsymbol{s})-V(\boldsymbol{s}) \\
&= 0
\end{aligned}
$$

∎

## A.2 Entropy dynamics under policy gradient

**Theorem 1.** *Given a policy gradient update $\widehat{\theta} := \theta + \alpha\cdot\nabla_\theta\mathcal{J}_{\mathrm{MDP}}(\boldsymbol{s})$, the expected change in entropy is approximately:*

$$
\Delta\mathcal{H}_{\pi_\theta}(\cdot\mid\boldsymbol{s}) \approx -\alpha\cdot\mathbb{E}_{a\sim\pi_\theta(\cdot|\boldsymbol{s}),a'\sim\pi_\theta(\cdot|\boldsymbol{s})}\left[A(\boldsymbol{s},a)\cdot L(\boldsymbol{s},a')\cdot u(\boldsymbol{s},a)^\top u(\boldsymbol{s},a')\right].
$$

$L(\boldsymbol{s},a) \overset{\mathrm{def}}{=} \log\pi_\theta(a\mid\boldsymbol{s}) - \mathbb{E}_{a\sim\pi_\theta(\cdot|\boldsymbol{s})}[\log\pi_\theta(a\mid\boldsymbol{s})]$ *denotes mean-centered log-probabilities and* $u(\boldsymbol{s},a) \overset{\mathrm{def}}{=} \nabla_\theta\log\pi_\theta(a\mid\boldsymbol{s})$ *is the score function for a policy $\pi_\theta$ evaluated at state $\boldsymbol{s}$ and action $a$.*

*Proof.* Let $L(\boldsymbol{s},a) \overset{\mathrm{def}}{=} \log\pi_\theta(a\mid\boldsymbol{s}) - \mathbb{E}_{a\sim\pi_\theta(\cdot|\boldsymbol{s})}[\log\pi_\theta(a\mid\boldsymbol{s})]$ denote mean-centered log-probabilities and let $u(\boldsymbol{s},a) \overset{\mathrm{def}}{=} \nabla_\theta\log\pi_\theta(a\mid\boldsymbol{s})$ denote the score function of policy $\pi_\theta$ evaluated at action $a$ and state $\boldsymbol{s}$. Let $g(\boldsymbol{s})$ and $h(\boldsymbol{s})$ denote the respective mean-baselined policy gradient and entropy gradient evaluated on-policy in some state $\boldsymbol{s}$:

$$
\begin{aligned}
g(\boldsymbol{s}) &= \nabla_\theta\mathcal{J}_{\mathrm{MDP}}(\boldsymbol{s}) = \mathbb{E}_{a\sim\pi_\theta(\cdot|\boldsymbol{s})}\left[A(\boldsymbol{s},a)\cdot u(\boldsymbol{s},a)\right] \\
h(\boldsymbol{s}) &= \nabla_\theta\mathcal{H}_{\pi_\theta}(\cdot\mid\boldsymbol{s}) = -\mathbb{E}_{a\sim\pi_\theta(\cdot|\boldsymbol{s})}\left[L(\boldsymbol{s},a)\cdot u(\boldsymbol{s},a)\right]
\end{aligned}
$$

Here, each estimator allows for an arbitrary baseline that cancels through the parameter gradient $\nabla_\theta$. While the baseline does not influence the exact mathematical construction, it does influence approximations to the change in entropy. Here we chose mean baselines to center the policy, minimize variance in each gradient estimator, and to agree with a tabular softmax approximation of the change in entropy (see Corollary 2).

Using the first-order Taylor approximation: $\mathcal{H}_{\pi_\theta}(\cdot \mid \boldsymbol{s} ; \theta + \alpha \cdot g) \approx \mathcal{H}_{\pi_\theta}(\cdot \mid \boldsymbol{s} ; \theta) + \alpha \cdot g^\top h$, for small learning rate $\alpha$, the expected change in entropy from a policy gradient update in state $\boldsymbol{s}$ is:

$$\Delta \mathcal{H}_{\pi_\theta}(\cdot \mid \boldsymbol{s}) \approx \alpha \cdot g(\boldsymbol{s})^\top h(\boldsymbol{s})$$
$$= -\alpha \cdot \left( \mathbb{E}_{a \sim \pi_\theta(\cdot \mid \boldsymbol{s})} \left[ A(\boldsymbol{s}, a) \cdot u(\boldsymbol{s}, a) \right] \right)^\top \left( \mathbb{E}_{a' \sim \pi_\theta(\cdot \mid \boldsymbol{s})} \left[ L(\boldsymbol{s}, a') \cdot u(\boldsymbol{s}, a') \right] \right)$$
$$= -\alpha \cdot \mathbb{E}_{a \sim \pi_\theta(\cdot \mid \boldsymbol{s}), a' \sim \pi_\theta(\cdot \mid \boldsymbol{s})} \left[ A(\boldsymbol{s}, a) \cdot L(\boldsymbol{s}, a') \cdot u(\boldsymbol{s}, a)^\top u(\boldsymbol{s}, a') \right]$$

∎

## A.3 Approximate entropy dynamics under policy gradient

**Corollary 1.** *Assuming $u(\boldsymbol{s}, a)^\top u(\boldsymbol{s}, a') = 0$ for all $a \neq a'$, the entropy change is proportional to:*
$$\Delta \mathcal{H}_{\pi_\theta}(\cdot \mid \boldsymbol{s}) \propto -\mathbb{E}_{a \sim \pi_\theta(\cdot \mid \boldsymbol{s})} \left[ A(\boldsymbol{s}, a) \cdot L(\boldsymbol{s}, a) \cdot \pi_\theta(a \mid \boldsymbol{s}) \right]$$

*Proof.* Assuming the score vectors satisfy orthogonality of the off-diagonal terms such that $u(\boldsymbol{s}, a)^\top u(\boldsymbol{s}, a') = 0$ for $a \neq a'$, the double expectation can be collapsed, yielding:
$$\Delta \mathcal{H}_{\pi_\theta}(\cdot \mid \boldsymbol{s}) \approx -\alpha \cdot \mathbb{E}_{a \sim \pi_\theta(\cdot \mid \boldsymbol{s})} \left[ \pi_\theta(a \mid \boldsymbol{s}) \cdot A(\boldsymbol{s}, a) \cdot L(\boldsymbol{s}, a) \cdot \|u(\boldsymbol{s}, a)\|^2 \right]$$

Assuming independence of the squared gradient norm magnitude, such that it can be treated as a constant with respect to the expectation,
$$\Delta \mathcal{H}_{\pi_\theta}(\cdot \mid \boldsymbol{s}) \propto -\mathbb{E}_{a \sim \pi_\theta(\cdot \mid \boldsymbol{s})} \left[ A(\boldsymbol{s}, a) \cdot L(\boldsymbol{s}, a) \cdot \pi_\theta(a \mid \boldsymbol{s}) \right]$$

∎

## A.4 Entropy dynamics under policy gradient for tabular softmax policies

**Proposition 1.** *For two functions $f(x)$ and $g(x)$ over samples $x \sim \pi_S$ of a softmax distribution $\pi_S(x) = \exp(S_x) / \sum_k \exp(S_k)$, the dot product of expected gradients is:*

$$\left\langle \mathbb{E}_{x \sim \pi_S}[f(x) \cdot \nabla_S \log \pi_S(x)] \quad , \quad \mathbb{E}_{y \sim \pi_S}[g(y) \cdot \nabla_S \log \pi_S(y)] \right\rangle = \mathbb{E}_{x \sim \pi_S}[\pi_S(x) \cdot (f(x) - \bar{f}) \cdot (g(x) - \bar{g})],$$

*where $\bar{f} = \mathbb{E}_{x \sim \pi_S}[f(x)]$ and $\bar{g} = \mathbb{E}_{x \sim \pi_S}[g(x)]$.*

*Proof.* First, let's compute $\nabla_S \log \pi_S(x)$ for the softmax distribution:

$$\log \pi_S(x) = \log \frac{\exp(S_x)}{\sum_k \exp(S_k)} = S_x - \log \sum_k \exp(S_k)$$

$$\nabla_{S_z} \log \pi_S(x) = \mathbb{1}_{x=z} - \frac{\exp(S_z)}{\sum_k \exp(S_k)} = \mathbb{1}_{x=z} - \pi_S(z)$$

where $\mathbb{1}_{x=y}$ is the indicator function (1 if $x = y$, 0 otherwise).

Now let's compute the dot product $\nabla_S \log \pi_S(x)^\top \nabla_S \log \pi_S(y)$:

$$\nabla_S \log \pi_S(x)^\top \nabla_S \log \pi_S(y) = \sum_j (\mathbb{1}_{x=j} - \pi_S(j))(\mathbb{1}_{y=j} - \pi_S(j))$$
$$= \sum_j (\mathbb{1}_{x=j} \cdot \mathbb{1}_{y=j} - \mathbb{1}_{x=j} \pi_S(j) - \pi_S(j) \cdot \mathbb{1}_{y=j} + \pi_S(j)^2)$$
$$= \mathbb{1}_{x=y} - \pi_S(x) - \pi_S(y) + \mathbb{E}_{z \sim \pi_S}[\pi_S(z)]$$

Now we can compute the dot product of expected gradients:

$$\left\langle \mathbb{E}_{x \sim \pi_S}[f(x) \cdot \nabla_S \log \pi_S(x)], \mathbb{E}_{y \sim \pi_S}[g(y) \cdot \nabla_S \log \pi_S(y)] \right\rangle$$
$$= \mathbb{E}_{x \sim \pi_S, y \sim \pi_S}[f(x) \cdot g(y) \cdot \nabla_S \log \pi_S(x)^\top \nabla_S \log \pi_S(y)]$$
$$= \mathbb{E}_{x \sim \pi_S, y \sim \pi_S}[f(x) \cdot g(y) \cdot (\mathbb{1}_{x=y} - \pi_S(x) - \pi_S(y) + \mathbb{E}_{z \sim \pi_S}[\pi_S(z)])]$$

Let's compute each term separately:

$$\mathbb{E}_{x\sim\pi_S, y\sim\pi_S}[f(x)\cdot g(y)\cdot\mathbb{1}_{x=y}] = \mathbb{E}_{x\sim\pi_S}[\pi_S(x)\cdot f(x)\cdot g(x)]$$
$$\mathbb{E}_{x\sim\pi_S, y\sim\pi_S}[f(x)\cdot g(y)\cdot\pi_S(x)] = \mathbb{E}_{x\sim\pi_S}[f(x)\cdot\pi_S(x)]\cdot\mathbb{E}_{y\sim\pi_S}[g(y)] = \mathbb{E}_{x\sim\pi_S}[\pi_S(x)\cdot f(x)]\cdot\bar{g}$$
$$\mathbb{E}_{x\sim\pi_S, y\sim\pi_S}[f(x)\cdot g(y)\cdot\pi_S(y)] = \mathbb{E}_{x\sim\pi_S}[f(x)]\cdot\mathbb{E}_{y\sim\pi_S}[g(y)\cdot\pi_S(y)] = \bar{f}\mathbb{E}_{y\sim\pi_S}[\pi_S(y)\cdot g(y)]$$
$$\mathbb{E}_{x\sim\pi_S, y\sim\pi_S}[f(x)\cdot g(y)\cdot\mathbb{E}_{z\sim\pi_S}[\pi_S(z)]] = \mathbb{E}_{z\sim\pi_S}[\pi_S(z)]\cdot\mathbb{E}_{x\sim\pi_S}[f(x)]\cdot\mathbb{E}_{y\sim\pi_S}[g(y)] = \mathbb{E}_{x\sim\pi_S}[\pi_S(x)]\cdot\bar{f}\cdot\bar{g}$$

Therefore:

$$\left\langle\mathbb{E}_{x\sim\pi_S}[f(x)\cdot\nabla_S\log\pi_S(x)], \mathbb{E}_{y\sim\pi_S}[g(y)\cdot\nabla_S\log\pi_S(y)]\right\rangle$$
$$= \mathbb{E}_{x\sim\pi_S}[\pi_S(x)\cdot f(x)\cdot g(x)] - \mathbb{E}_{x\sim\pi_S}[\pi_S(x)\cdot f(x)]\cdot\bar{g} - \bar{f}\cdot\mathbb{E}_{x\sim\pi_S}[\pi_S(x)\cdot g(x)] + \mathbb{E}_{x\sim\pi_S}[\pi_S(x)]\cdot\bar{f}\cdot\bar{g}$$
$$= \mathbb{E}_{x\sim\pi_S}[\pi_S(x)\cdot(f(x)\cdot g(x) - f(x)\cdot\bar{g} - \bar{f}\cdot g(x) + \bar{f}\cdot\bar{g})]$$
$$= \mathbb{E}_{x\sim\pi_S}[\pi_S(x)\cdot(f(x) - \bar{f})\cdot(g(x) - \bar{g})]$$

where $\bar{f} = \mathbb{E}_{x\sim\pi_S}[f(x)]$ and $\bar{g} = \mathbb{E}_{x\sim\pi_S}[g(x)]$. ∎

The above proposition holds for simple softmax policies, but involves a much more complex gradient term and inner product for generic transformer-based policies.

**Corollary 2.** *Under a tabular softmax policy, a policy gradient update $\widehat{\theta} := \theta + \alpha\cdot\nabla_\theta\mathcal{J}_{\mathrm{MDP}}$ changes the entropy approximately:*

$$\Delta\mathcal{H}_{\pi_\theta}(\cdot\mid s) \approx -\alpha\cdot\mathbb{E}_{a\sim\pi_S(\cdot\mid s)}\left[\pi_S(a\mid s)\cdot\left(\log\pi_S(a\mid s) - \overline{\log\pi_S(\cdot\mid s)}\right)\cdot\left(R(s,a) - \overline{R(s)}\right)\right]$$

*where $\overline{\log\pi_S(\cdot\mid s)} = \mathbb{E}_{a\sim\pi_\theta(\cdot\mid s)}\left[\log\pi_S(a\mid s)\right]$ and $\overline{R(s)} = \mathbb{E}_{a\sim\pi_\theta(\cdot\mid s)}\left[R(s,a)\right]$.*

*Proof.* Let $g(s)$ and $h(s)$ denote the respective policy gradient and entropy gradient evaluated on-policy in some state $s$:

$$g(s) = \nabla_\theta\mathcal{J}_{\mathrm{MDP}}(s) = \mathbb{E}_{a\sim\pi_\theta(\cdot\mid s)}\left[R(s,a)\cdot\nabla_\theta\log\pi_\theta(a\mid s)\right]$$
$$h(s) = \nabla_\theta\mathcal{H}_{\pi_\theta}(\cdot\mid s) = -\mathbb{E}_{a\sim\pi_\theta(\cdot\mid s)}\left[\log\pi_\theta(a\mid s)\cdot\nabla_\theta\log\pi_\theta(a\mid s)\right]$$

Using the first-order Taylor approximation: $\mathcal{H}_{\pi_\theta}(\cdot\mid s\,;\,\theta+\alpha\cdot g) \approx \mathcal{H}_{\pi_\theta}(\cdot\mid s\,;\,\theta) + \alpha\cdot g^\top h$, for small learning rate $\alpha$, the expected change in entropy from a policy gradient update in state $s$ is:

$$\Delta\mathcal{H}_{\pi_\theta}(\cdot\mid s) \approx \alpha\cdot g(s)^\top h(s)$$
$$= -\alpha\cdot\mathbb{E}_{a\sim\pi_S(\cdot\mid s)}\left[\pi_S(a\mid s)\cdot\left(\log\pi_S(a\mid s) - \overline{\log\pi_S(\cdot\mid s)}\right)\cdot\left(R(s,a) - \overline{R(s)}\right)\right]$$

The second line follows Prop. 1. Note that gradient interactions through the softmax automatically center the reward function, i.e., $A(s,a) = R(s,a) - V(s) = R(s,a) - \overline{R(s)}$. The log-probabilities, too, are centered as here they reflect $R(s,a) = -\log\pi_S(a\mid s)$. This yields a form equivalent to Corollary 1. ∎

## A.5 ENTROPY DYNAMICS UNDER CLIPPED PPO

**Proposition 2.** *Given two distributions $\pi(x)$ and $\phi(x)$ with constraint $\frac{\pi(x)}{\phi(x)} \leq 1+\epsilon$ for all $x$, their relative entropy is bound by*

$$\mathcal{H}(\pi) \leq (1+\epsilon)\cdot\mathcal{H}(\phi)$$

*Proof.* Let's parametrize $\pi(x) = \beta_x \phi(x)$ with $\beta_x \geq 0$ and compute its probability

$$
\begin{aligned}
\mathcal{H}(\pi) &= -\mathbb{E}_{x \sim \pi}\left[\log \pi(x)\right] \\
&= -\mathbb{E}_{x \sim \pi}\left[\log \phi(x)\right] - \mathbb{E}_{x \sim \pi}\left[\log \beta_x\right] \\
&= -\mathbb{E}_{x \sim \pi}\left[\log \phi(x)\right] - \underbrace{\mathbb{E}_{x \sim \pi}\left[\log \frac{\pi(x)}{\phi(x)}\right]}_{D_{\mathrm{KL}}(\pi \| \phi) \geq 0} \\
&\leq -\mathbb{E}_{x \sim \pi}\left[\log \phi(x)\right] \\
&= -\mathbb{E}_{x \sim \phi}\left[\frac{\pi(x)}{\phi(x)} \log \phi(x)\right] \\
&= \mathbb{E}_{x \sim \phi}\left[\beta_x \cdot -\log \phi(x)\right] \\
&\leq \mathbb{E}_{x \sim \phi}\left[(1 + \epsilon) \cdot -\log \phi(x)\right] \\
&= (1 + \epsilon) \cdot \mathcal{H}(\phi)
\end{aligned}
$$

The second-last line uses $-\log \phi(x) \geq 0$ and $\beta_x \leq (1 + \epsilon)$ by definition, hence $\beta_x \cdot -\log \phi(x) \leq (1 + \epsilon) \cdot -\log \phi(x)$. ∎

**Theorem 2.** *Proximal Policy Optimization (PPO) bounds the entropy $\mathcal{H}_{\pi_{\theta^{new}}}(\cdot \mid s)$ of the updated policy by the original policy entropy $\mathcal{H}_{\pi_{\theta^{old}}}(\cdot \mid s)$ such that:*

$$
(1 - \epsilon_{low}) \cdot \mathcal{H}_{\pi_{\theta^{old}}}(\cdot \mid s) \leq \mathcal{H}_{\pi_{\theta^{new}}}(\cdot \mid s) \leq (1 + \epsilon_{high}) \cdot \mathcal{H}_{\pi_{\theta^{old}}}(\cdot \mid s)
$$

*Proof.* Applying Prop. 2 to $\frac{\pi_\theta^{new}}{\pi_\theta^{old}} \leq 1 + \epsilon_{\text{high}}$ yields the upper bound

$$
\mathcal{H}_{\pi_{\theta^{new}}}(\cdot \mid s) \leq (1 + \epsilon_{\text{high}}) \cdot \mathcal{H}_{\pi_{\theta^{old}}}(\cdot \mid s).
$$

Applying Prop. 2 to $1 - \epsilon_{\text{low}} \leq \frac{\pi_\theta^{new}}{\pi_\theta^{old}}$ (equivalently $\frac{\pi_\theta^{old}}{\pi_\theta^{new}} \leq \frac{1}{1-\epsilon_{\text{low}}}$) yields the lower bound

$$
\mathcal{H}_{\pi_{\theta^{old}}}(\cdot \mid s) \leq \frac{1}{1 - \epsilon_{\text{low}}} \cdot \mathcal{H}_{\pi_{\theta^{new}}}(\cdot \mid s)
$$

or equivalently

$$
(1 - \epsilon_{\text{low}}) \cdot \mathcal{H}_{\pi_{\theta^{old}}}(\cdot \mid s) \leq \mathcal{H}_{\pi_{\theta^{new}}}(\cdot \mid s).
$$

∎

## A.6 Entropy change under $A_{\text{REPO}}$ advantage function

**Proposition 3.** *For advantage $A_{\text{REPO}}(s, a) \stackrel{\text{def}}{=} A(s, a) - \beta_s \cdot L(s, a)$, the first–order change in entropy induced by a policy–gradient step is:*

$$
\Delta \mathcal{H}_{\pi_\theta}^{\text{REPO}}(\cdot \mid s) \approx \Delta \mathcal{H}_{\pi_\theta}(\cdot \mid s) + \beta_s \cdot \alpha \cdot \left\| \mathbb{E}_{a \sim \pi_\theta(\cdot \mid s)}\left[L(s, a) \cdot u(s, a)\right] \right\|^2.
$$

*Proof.* Let $g(s) = \mathbb{E}_{a \sim \pi_\theta(\cdot \mid s)}\left[A(s, a) \cdot u(s, a)\right]$ and $h(s) = -\mathbb{E}_{a \sim \pi_\theta(\cdot \mid s)}\left[L(s, a) \cdot u(s, a)\right]$ denote the respective policy gradient and entropy gradient evaluated on-policy in some state $s$.

Using $A_{\text{REPO}}$, the policy gradient becomes:

$$
g_{\text{REPO}}(s) = \mathbb{E}_{a \sim \pi_\theta(\cdot \mid s)}\left[(A(s, a) - \beta_s L(s, a)) \cdot u(s, a)\right]
$$

The first–order entropy change is:

$$
\begin{aligned}
\Delta \mathcal{H}_{\pi_\theta}^{\text{REPO}}(\cdot \mid s) &\approx \alpha \cdot g_{\text{REPO}}(s)^\top h(s) \\
&= \alpha \cdot \left(\mathbb{E}_{a \sim \pi_\theta(\cdot \mid s)}\left[(A(s, a) - \beta_s L(s, a)) \cdot u(s, a)\right]\right)^\top h(s) \\
&= \alpha \cdot \left(\mathbb{E}_{a \sim \pi_\theta(\cdot \mid s)}\left[A(s, a) \cdot u(s, a)\right]\right)^\top h(s) - \beta_s \cdot \alpha \cdot \left(\mathbb{E}_{a \sim \pi_\theta(\cdot \mid s)}\left[L(s, a) \cdot u(s, a)\right]\right)^\top h(s) \\
&= \alpha \cdot g(s)^\top h(s) + \beta_s \cdot \alpha \cdot h(s)^\top h(s) \\
&= \Delta \mathcal{H}_{\pi_\theta}(\cdot \mid s) + \beta_s \cdot \alpha \cdot \left\| \mathbb{E}_{a \sim \pi_\theta(\cdot \mid s)}\left[L(s, a) \cdot u(s, a)\right] \right\|^2.
\end{aligned}
$$

∎

A.7  CONNECTION TO ENTROPY REGULARIZATION

The REPO objective has a direct connection to standard entropy regularization. Consider the REPO-D advantage for a given state $s$:

$$A_{\mathrm{REPO}}(s, a) = A(s, a) - \beta_s \cdot L(s, a).$$

The resulting policy gradient is:

$$
\nabla_\theta \mathcal{J}_{\mathrm{REPO}}(s) = \mathbb{E}_{a \sim \pi_\theta(\cdot|s)} \left[ A_{\mathrm{REPO}}(s, a) \cdot u(s, a) \right]
$$
$$
= \underbrace{\mathbb{E}_{a \sim \pi_\theta(\cdot|s)} \left[ A(s, a) \cdot u(s, a) \right]}_{\nabla_\theta \mathcal{J}_{\mathrm{MDP}}(s)} + \beta_s \cdot \underbrace{\mathbb{E}_{a \sim \pi_\theta(\cdot|s)} \left[ -L(s, a) \cdot u(s, a) \right]}_{\nabla_\theta \mathcal{H}_{\pi_\theta}(\cdot|s)},
$$

where the second equality follows from Lemma 5. Thus, REPO-D is *exactly equivalent* to augmenting the MDP objective with an explicit entropy bonus:

$$\mathcal{J}_{\mathrm{REPO}}(s) = \mathcal{J}_{\mathrm{MDP}}(s) + \beta_s \cdot \mathcal{H}_{\pi_\theta}(\cdot \mid s).$$

**Advantage over an explicit entropy bonus.**  Despite this equivalence, REPO-D offers two practical advantages over directly computing and differentiating the entropy bonus.

**Zero additional memory cost.** Computing an exact entropy bonus requires materializing the full logit vector over the vocabulary for every token in every trajectory, as the entropy $\mathcal{H}_{\pi_\theta}(\cdot \mid s) = -\sum_a \pi_\theta(a \mid s) \log \pi_\theta(a \mid s)$ is a sum over all vocabulary entries. For large vocabularies (e.g., $> 10^5$ tokens) and long trajectories, this is unnecessarily memory-intensive. In contrast, REPO-D estimates the entropy gradient via REINFORCE using only the log-probability of the *sampled* token, which is already computed during the forward pass when using Cut Cross-Entropy (Wijmans et al., 2025). REPO-D therefore incurs **zero additional memory cost** beyond the standard policy gradient computation.

**Variance reduction via control variates.** Because REPO-D couples the advantage $A(s, a)$ and the centered log-probability $L(s, a)$ at the level of individual sampled actions, it naturally acts as a *control variate* for the policy gradient estimator. When advantages and log-probabilities are positively correlated—as is typical during the entropy-collapsing phase of training, per Corollary 1—subtracting the $\beta_s \cdot L$ term reduces variance in the combined gradient estimate. This coupling would be lost if the policy gradient and entropy bonus were estimated from independent samples.

A.8  PROOF OF THEOREM 3 (BF16 MULTIPLICATIVE BIAS)

**Theorem 3.** *Under bf16 quantization, the observed ratio exhibits a multiplicative upward bias:* $\mathbb{E}[r_{observed} \mid r_{true}] > r_{true}$. *[Proof in App. A.8].*

*Proof.* Let $\varepsilon_{\mathrm{new}}$ and $\varepsilon_{\mathrm{old}}$ denote the quantization errors:

$$\log \pi_\theta(a|s) = \mathrm{bf16}(\log \pi_\theta(a|s)) + \varepsilon_{\mathrm{new}} \tag{1}$$

$$\log \pi_{\theta_{\mathrm{old}}}(a|s) = \mathrm{bf16}(\log \pi_{\theta_{\mathrm{old}}}(a|s)) + \varepsilon_{\mathrm{old}} \tag{2}$$

Assuming the lower 16 mantissa bits are uniformly distributed, $\varepsilon_{\mathrm{new}}, \varepsilon_{\mathrm{old}} \sim$ Uniform$(-\mathrm{ulp}/2, \mathrm{ulp}/2)$ independently, where ulp is the unit-in-last-place at the relevant exponent.

The true ratio can be expressed as:

$$r_{\mathrm{true}} = \exp(\mathrm{bf16}(\log \pi_\theta) - \mathrm{bf16}(\log \pi_{\theta_{\mathrm{old}}}) + \varepsilon_{\mathrm{new}} - \varepsilon_{\mathrm{old}}) \tag{3}$$

$$= r_{\mathrm{observed}} \cdot \exp(\delta) \tag{4}$$

where $\delta = \varepsilon_{\mathrm{new}} - \varepsilon_{\mathrm{old}}$.

Therefore: $r_{\mathrm{observed}} = r_{\mathrm{true}} \cdot \exp(-\delta)$

Since $\delta$ is symmetric around zero with $\mathrm{Var}(\delta) = \mathrm{Var}(\varepsilon_{\mathrm{new}}) + \mathrm{Var}(\varepsilon_{\mathrm{old}}) = (\mathrm{ulp}_{\mathrm{new}}^2 + \mathrm{ulp}_{\mathrm{old}}^2)/12$, Jensen's inequality for the convex function $\exp(-\cdot)$ yields:

$$\mathbb{E}[r_{\mathrm{observed}} \mid r_{\mathrm{true}}] = r_{\mathrm{true}} \cdot \mathbb{E}[\exp(-\delta)] \tag{5}$$

$$> r_{\mathrm{true}} \cdot \exp(\mathbb{E}[-\delta]) \tag{6}$$

$$= r_{\mathrm{true}} \tag{7}$$

For small quantization errors, a Taylor expansion gives:

$$\mathbb{E}[r_{\text{observed}} \mid r_{\text{true}}] \approx r_{\text{true}} \cdot (1 + \text{Var}(\delta)/2) \tag{8}$$

$$= r_{\text{true}} \cdot (1 + (\text{ulp}_{\text{new}}^2 + \text{ulp}_{\text{old}}^2)/24) \tag{9}$$

The bias is **proportional to** $r_{\text{true}}$: larger ratios experience proportionally larger absolute bias.  ∎

# B    NUMERICAL CONSIDERATIONS: ADDITIONAL DETAILS

The main findings on numerical considerations (FSDP2 output casting, FP16 vs BF16 training, and their impact on entropy dynamics) are presented in §4. This appendix provides additional technical details.

## B.1    ADDITIONAL DETAILS ON FSDP2 OUTPUT CASTING

As described in App. C.2, we use the FSDP2 framework for distributed training on multiple GPUs (Zhang et al., 2024). In the HuggingFace *Accelerate* library, FSDP2 is configured to cast all module outputs to the chosen floating-point type (e.g., BF16), including the final model outputs, even when the computations involving logits (such as softmax) are performed in full 32-bit precision.

This is the default behavior of the library, and at the time of submission there was no single configuration parameter to switch it off. To preserve full-precision log probabilities, the user must explicitly override the `output_dtype` of the `MixedPrecisionPolicy` (MPP) object (see `torch/distributed/fsdp/_fully_shard/_fsdp_api.py` for details).

Naively, this cast should not affect the RL gradients, as the backward pass of such a casting operation is the identity function. Indeed, there appears to be no measurable difference for fully on-policy algorithms like RLOO. The half-precision downcast, however, does measurably impact the numerical stability of the importance weight and thus can affect off-policy algorithms that use clipping, such as LOOP, GRPO, and DAPO.

Fig. 2a empirically demonstrates the clipping bias introduced by the 16-bit rounding when training with DAPO. We observe that when the rounding is present (before the `MixedPrecisionPolicy` fix), more tokens get clipped due to exceeding the higher end of the range $\epsilon_{high}$ preventing probability increase for low probability tokens and thus reducing overall entropy. At the same time, fewer tokens are clipped due to $\epsilon_{low}$. The overall effect is the tightening of the clipping on the higher end of the range while relaxing it on the lower end, resulting in the reduced effectiveness of entropy preservation from the asymmetric clipping. It can be further noted that the 16-bit rounding changes the clipping outcome only for a tiny fraction of tokens, fewer than 0.1% of the total number of output tokens. This suggests that a very small number of pivotal tokens play an essential role in learning and warrants further study of this effect.

App. B.3 empirically confirms the significant impact of half-precision rounding on the overall performance and entropy dynamics (see Fig. 8).

## B.2    FLOAT16 TRAINING

In our original experiments, the models were trained exclusively in *bfloat16* (BF16), which has become common practice in LLM training because of its higher dynamic range. Recent publications (Qi et al., 2025) reported improved RL training with *float16* (FP16) floating-point format as its additional 3 mantissa bits enable more accurate gradient representation.

In addition, the choice of floating-point format affects the discrepancy between inference (vLLM) and training policies. These discrepancies are inherent to RL systems with a separate inference server and arise from small differences in model-layer implementations as well as from the lack of batch-size invariance in GPU kernels. In our experiments, we find that FP16 training significantly reduces the inference-training discrepancy (see Fig. 2b).

## B.3    ABLATION STUDY

Fig. 8 summarizes the ablation study of the numerical tweaks described in Apps. B.1 and B.2 performed for DAPO training on `Qwen-3-8B`. We observe that when the MPP fix and FP16 training are used together, the entropy dynamics of DAPO change completely, from collapse and sub-par exploration to a rapid increase in entropy over the course of training. More generally, we observed improved training across models and algorithm variants when both of the above changes were applied.

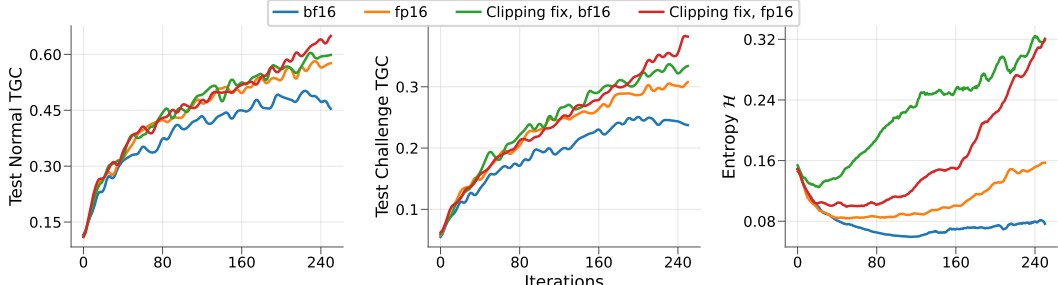

Figure 8: Cumulative effect of the MixedPrecisionPolicy (MPP) fix and FP16 training when applied to DAPO algorithm with `Qwen-3-8B`. Each curve represents the mean of three independent runs (seeds). This is the expanded version of Fig. 3.

### B.4 SOFTMAX GRADIENT NUMERICAL PRECISION

An additional numerical issue arises in the backward pass of the softmax function when computing gradients for high-probability tokens. For a token with probability $p$, the gradient computation involves the term $-A \cdot (1-p)$ for the sampled token and $A \cdot p$ for all other tokens in the vocabulary. When $p$ exceeds approximately $1 - 2^{-23} \approx 0.9999999$ in single precision (or the corresponding threshold in half precision), the value $1 - p$ rounds to exactly zero, causing the gradient to vanish entirely for high-confidence sampled tokens while the gradient for all other tokens is still non-zero. Essentially this makes the gradient vectors for high-confidence tokens incorrect. Surprisingly, in our experiments this affects on the order of 30-40% of tokens; these high-confidence tokens often correspond to chat template formatting or programming language syntax.

This issue is distinct from the BF16 multiplicative bias discussed in App. B.1 and affects even full-precision training when the model becomes highly confident. The problem manifests as:

- Gradients for high-probability tokens become incorrect
- Learning dynamics may be distorted for tokens where the model is already very confident

**Fix.** The solution involves an $O(1)$ per time step recalculation of the problematic terms in 64-bit precision. Specifically, when computing the softmax gradient, we identify the highest probability tokens for each time step and simply recalculate their gradient in softmax backward step using 64-bit arithmetic. This targeted approach preserves numerical accuracy for high-confidence tokens without incurring the memory overhead of full 64-bit gradient computation for the entire vocabulary.

In practice, this effect is much less noticeable compared to the other numerical improvements (FP16 training and MixedPrecisionPolicy fix) mostly because the magnitude of affected gradients is tiny compared to tokens with $p \ll 1$. The combined effect of all numerical fixes is shown in Fig. 8.

## C  ADDITIONAL EXPERIMENT DETAILS

### C.1  ENVIRONMENTS

*Interactive tool-use agent.* For the AppWorld benchmark, rollouts proceed in turns (up to 30 turns during training and 50 during evaluation), in a manner akin to an interactive notebook.

During each turn, the model generations are parsed to extract any Python code blocks, potentially containing calls to AppWorld API. These are executed to retrieve information or alter the environment state. The outputs of successful API calls or the error trace of incorrect calls appear in the agent's context after each turn. Once done, the agent may mark a task as completed at which point it is assessed whether the task state was updated successfully. Failure to mark the task as complete within the turn limit or context limit (32K) results in a failure. Sparse outcome-based rewards in $[0, 1]$ are assigned during training as the fraction of passing unit-tests. Binary rewards in $\{0, 1\}$ are used during evaluation requiring complete correctness.

*Mathematical reasoning.* For the AIME benchmarks, model responses are processed and scored using the Eleuther AI lm-eval-harness Minerva math parsing utilities (Gao et al., 2024). The final unnormalized answer is first identified and parsed, then the answer is normalized to remove units, formatting, etc., and finally equivalence between the model answer and reference answer is determined using Sympy (Meurer et al., 2017).

### C.2  TRAINING

Experiments are typically executed on 3 NVIDIA H100 8-GPU nodes. One node is used for rollout generation, one for learning, and one for evaluation. Rollouts are generated using two instances of vLLM (Kwon et al., 2023) servers using 4 GPUs each with tensor parallelism. Custom RL implementation based on FSDP2 (Zhang et al., 2024) is used for training. To account for any discrepancies between sampling and training subsystems, the log-probabilities of rollout tokens are recalculated on the training node to ensure accurate importance weights for backpropagation. Cut-Cross-Entropy (CCE) is used to reduce the memory footprint during training by preventing the materialization of all logits except the target (Wijmans et al., 2025). Models are fine-tuned with LoRA ($rank = 16$, $\alpha = 32$) on the self-attention (key, value, query, output) and MLP modules (Hu et al., 2022). We use an `AdamW` optimizer with a constant learning rate of $5 \times 10^{-5}$, `weight-decay` $= 0.01$, and gradient clipping with `max-norm` $= 0.1$. To speed up rollout collection, we introduce an early stopping criteria. Once at least 4/6 rollouts per task and 90% of total rollouts are collected, we immediately proceed to training to prevent bottlenecks caused by very few extra long generations (Wijmans et al., 2020; Chen et al., 2025a).

### C.3  SUMMARY OF RESULTS

This section provides additional analysis of the performance of different algorithmic variants, see Tabs. 1 to 4.

Each row in the table corresponds to three independent training runs. REPO-R is implemented on top of GRPO unless stated otherwise (since GRPO experiences significant exploration collapse without entropy regulation).

Entropy bonus baseline is implemented on top of GRPO with the adaptive coefficient described in §5.1. Due to higher memory usage, these experiments with `Qwen-3-32B` required B200 training nodes.

For AppWorld, in each run we find the checkpoint with the highest validation score (based on *dev* dataset split) and report the mean scores on test splits across independent seeds, standard deviation, as well as the best score on a particular test split across all seeds. For AIME we simply pick the checkpoint with the highest average test score between AIME '24 and '25 since our setup lacks a separate validation split. The last column reports the change in entropy (absolute and relative) for the best checkpoint compared to the initial policy.

Tabs. 1 and 2 show strong performance of entropy-preserving methods compared to baselines (DAPO and GRPO respectively), as well as more stable entropy dynamics.

On AIME (Tabs. 3 and 4), ADAPO and REPO-R perform competitively with other off-policy methods, however the results are very close between all algorithms and the peak performance is unlocked very early in training (Fig. 6). This suggests that the base models might already be overfit to this benchmark. To make the task harder and to make the training dynamics more interesting, we limit the maximum allowed context length to 4096 tokens, requiring the models to learn more efficient and compact reasoning.

With numerical improvements described in Apps. B.2 and B.4, on-policy RLOO performs remarkably well in all our tests despite slower progress in early training iterations. We reach 79% success rate on Test Normal and 71% on Test Challenge, significantly exceeding the highest reported scores at the time of submission (`https://appworld.dev/leaderboard`) achieved with an agentic GPT-4.1-based system (Marreed et al., 2025).

| Algorithm | Test Normal | Best TN | Test Challenge | Best TC | $\Delta\mathcal{H}$ |
|---|---|---|---|---|---|
| RLOO | $0.53 \pm 0.05$ | **0.64** | $0.32 \pm 0.05$ | **0.40** | +0.01 (+4%) |
| GRPO | $0.46 \pm 0.01$ | 0.46 | $0.21 \pm 0.02$ | 0.23 | -0.09 (-64%) |
| LOOP | $0.40 \pm 0.02$ | 0.42 | $0.19 \pm 0.02$ | 0.22 | -0.08 (-53%) |
| GSPO | $\mathbf{0.56 \pm 0.04}$ | 0.60 | $\mathbf{0.32 \pm 0.03}$ | 0.36 | +0.00 (+2%) |
| DAPO | $0.51 \pm 0.03$ | 0.55 | $0.28 \pm 0.01$ | 0.29 | +0.06 (+40%) |
| GRPO + $\mathcal{H}$ bonus | $0.48 \pm 0.03$ | 0.52 | $0.26 \pm 0.02$ | 0.28 | -0.01 (-6%) |
| ADAPO | $0.53 \pm 0.02$ | 0.57 | $0.30 \pm 0.02$ | 0.33 | -0.00 (-2%) |
| REPO-R | $0.49 \pm 0.01$ | 0.49 | $0.25 \pm 0.02$ | 0.27 | -0.01 (-9%) |

Table 1: Task goal completion scores for AppWorld with `Qwen-3-8B` by training algorithm.

| Algorithm | Test Normal | Best TN | Test Challenge | Best TC | $\Delta\mathcal{H}$ |
|---|---|---|---|---|---|
| RLOO | $0.77 \pm 0.02$ | 0.79 | $\mathbf{0.61 \pm 0.06}$ | **0.71** | -0.18 (-36%) |
| GRPO | $0.67 \pm 0.03$ | 0.71 | $0.46 \pm 0.03$ | 0.49 | -0.29 (-57%) |
| LOOP | $0.66 \pm 0.02$ | 0.68 | $0.45 \pm 0.03$ | 0.47 | -0.31 (-60%) |
| GSPO | $0.69 \pm 0.01$ | 0.70 | $0.51 \pm 0.00$ | 0.51 | -0.07 (-14%) |
| DAPO | $0.73 \pm 0.04$ | 0.77 | $0.52 \pm 0.02$ | 0.55 | +1.52 (+298%) |
| GRPO + $\mathcal{H}$ bonus | $0.71 \pm 0.02$ | 0.74 | $0.49 \pm 0.02$ | 0.51 | +0.14 (+27%) |
| ADAPO | $\mathbf{0.78 \pm 0.02}$ | **0.82** | $0.58 \pm 0.03$ | 0.62 | +0.52 (+102%) |
| REPO-R | $0.73 \pm 0.04$ | 0.78 | $0.54 \pm 0.06$ | 0.63 | +0.03 (+7%) |

Table 2: Task goal completion scores for AppWorld with `Qwen-3-32B` by training algorithm.

| Algorithm | AIME 2024 | Best '24 | AIME 2025 | Best '25 | $\Delta\mathcal{H}$ |
|---|---|---|---|---|---|
| RLOO | $\mathbf{0.47 \pm 0.01}$ | **0.48** | $0.33 \pm 0.01$ | 0.34 | -0.04 (-29%) |
| GRPO | $0.45 \pm 0.02$ | 0.47 | $0.32 \pm 0.02$ | 0.35 | -0.07 (-47%) |
| GSPO | $0.43 \pm 0.03$ | 0.47 | $0.34 \pm 0.01$ | 0.34 | +0.05 (+37%) |
| DAPO | $0.43 \pm 0.01$ | 0.44 | $0.32 \pm 0.02$ | 0.33 | +0.23 (+158%) |
| ADAPO | $0.43 \pm 0.00$ | 0.43 | $0.34 \pm 0.02$ | **0.36** | +0.13 (+93%) |
| REPO-R | $0.45 \pm 0.01$ | 0.46 | $\mathbf{0.34 \pm 0.01}$ | 0.35 | +0.13 (+91%) |

Table 3: Scores for AIME with `Qwen-3-8B` by training algorithm. The model is limited to 4K context length.

| Algorithm | AIME 2024 | Best '24 | AIME 2025 | Best '25 | $\Delta\mathcal{H}$ |
|---|---|---|---|---|---|
| RLOO | **0.53** ± 0.01 | **0.55** | **0.39** ± 0.01 | **0.40** | +0.04 (+28%) |
| GRPO | 0.51 ± 0.01 | 0.52 | 0.36 ± 0.02 | 0.39 | +0.06 (+40%) |
| DAPO | 0.43 ± 0.01 | 0.44 | 0.31 ± 0.03 | 0.35 | +0.75 (+519%) |
| ADAPO | 0.46 ± 0.03 | 0.50 | 0.34 ± 0.01 | 0.35 | +0.29 (+198%) |
| REPO-R | 0.49 ± 0.02 | 0.51 | 0.35 ± 0.02 | 0.38 | +0.17 (+117%) |

Table 4: Scores for AIME with `Qwen-3-32B` by training algorithm. The model is limited to 4K context length.

# D  ADDITIONAL RESULTS

## D.1  GEOMETRIC INTERPRETATION OF REPO

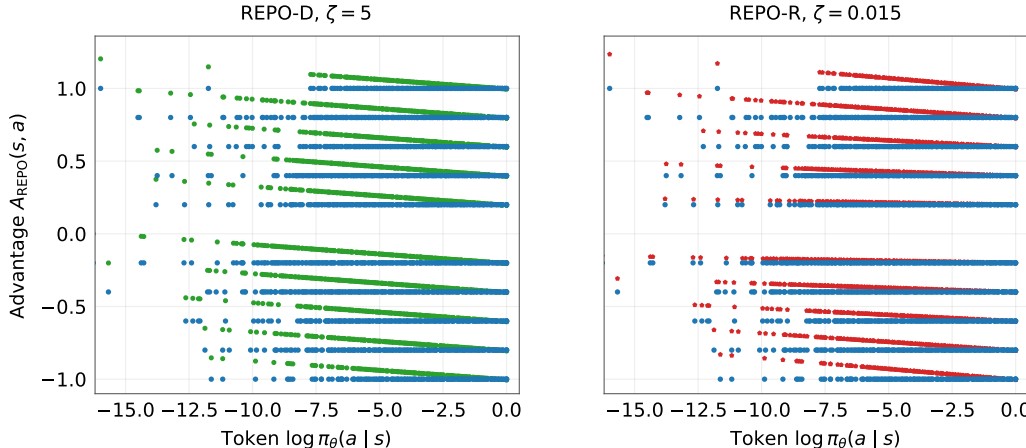

Figure 9: The REPO transformation rotates $(A, \log \pi)$ pairs, promoting low probability actions. Original unmodified advantages shown in blue, advantages $A_{\text{REPO-D}}$ are shown in green and $A_{\text{REPO-R}}$ in red. This plot demonstrates the transformation in a `Qwen-3-32B` AIME experiment, showcasing the real distribution of the data: $(A, \log \pi)$ pairs are highly concentrated near $\log \pi \approx 0.0$.

The transformation induced by REPO algorithm can be viewed in Fig. 9. REPO-D reflects a consistent rotation across the space, boosting the advantages of actions proportional to their surprisals ($-\log \pi_\theta$). REPO-R scales advantages proportionally not only to the surprisal, but to the magnitude of the advantage. Intuitively, this strongly reinforces low-probability correct actions, especially when they yield outcomes significantly better than average for a given batch of experience.

Fig. 9 uses data from the `Qwen-3-32B` AIME experiment. The parameters of the algorithm are revealed in the structure of the data: there are 5 distinct positive and negative advantage values, corresponding to 5 unique outcomes of group-based advantage estimation (1 success / 5 failures, 2 successes / 4 failures, etc.). Groups with zero advantages are filtered out.

With the appropriate value of $\zeta$, REPO-D transformation counteracts the covariance-like term in $\Delta \mathcal{H}$ approximation, therefore REPO-D is short for REPO-Decorrelate. REPO-R is a shorthand for REPO-Rescale, as it simply rescales the advantages by a multiplicative factor $1 \pm \zeta \cdot L(\boldsymbol{s}, a)$.

## D.2  REPO-R: DERIVATION AND IMPLEMENTATION DETAILS

We derive the practical REPO-R formula from the general REPO advantage and discuss the simplifications made in practice.

**Derivation.**  Starting from the general REPO advantage (§5.1):

$$A_{\text{REPO}}(\boldsymbol{s}, a) = A(\boldsymbol{s}, a) - \beta_{\boldsymbol{s}} \cdot L(\boldsymbol{s}, a),$$

REPO-R sets $\beta_{\boldsymbol{s},a}^{\text{REPO-R}} = \zeta \cdot |A(\boldsymbol{s}, a)|$, yielding:

$$A_{\text{REPO-R}}(\boldsymbol{s}, a) = A(\boldsymbol{s}, a) - \zeta \cdot |A(\boldsymbol{s}, a)| \cdot L(\boldsymbol{s}, a).$$

For positive advantages ($A > 0$), this simplifies to:

$$A_{\text{REPO-R}}^{+}(\boldsymbol{s}, a) = A(\boldsymbol{s}, a) \cdot (1 - \zeta \cdot L(\boldsymbol{s}, a)).$$

Since $L(\boldsymbol{s}, a) = \log \pi_\theta(a \mid \boldsymbol{s}) - \overline{\log \pi_\theta(\cdot \mid \boldsymbol{s})}$, this boosts rare correct actions (which have more negative $L$) and attenuates common ones when $\zeta$ is positive. Negative value of $\zeta$ reverses the effect punishing exceedingly rare tokens.

For negative advantages ($A < 0$), we have:

$$A_{\text{REPO-R}}^-(\boldsymbol{s}, a) = A(\boldsymbol{s}, a) \cdot (1 + \zeta \cdot L(\boldsymbol{s}, a)).$$

Because $L(\boldsymbol{s}, a)$ is negative for rare actions and positive for common ones, the factor $(1 + \zeta \cdot L)$ decreases penalties on rare incorrect actions and strengthens penalties on common incorrect actions (for $\zeta > 0$).

**Practical simplification: using $\log \pi_\theta$ instead of centered $L$.** The theoretically motivated formulation uses the centered log-probability $L(\boldsymbol{s}, a) = \log \pi_\theta(a \mid \boldsymbol{s}) - \mathbb{E}_{a' \sim \pi_\theta(\cdot | \boldsymbol{s})}[\log \pi_\theta(a' \mid \boldsymbol{s})]$. In practice, we replace $L(\boldsymbol{s}, a)$ with the raw log-probability $\log \pi_\theta(a \mid \boldsymbol{s})$. An implementation using $L(\boldsymbol{s}, a)$ would require recomputing $\overline{\log \pi_\theta(\cdot \mid \boldsymbol{s})}$ over the entire dataset of trajectories after each gradient step, which is computationally expensive, and omitting it has negligible effect in practice.

**Reference implementation.** The following code listing shows the REPO-R advantage rescaling and the adaptive controller used in our experiments.

```python
# Adaptive controller (once per iteration)
if current_entropy > target_entropy:
    if zeta >= 0:
        zeta /= 2.0
        if zeta < zeta_min:
            zeta = -zeta_min  # flip the sign if needed
    else:
        zeta = max(-zeta_max, zeta * 2)
elif current_entropy < target_entropy:
    if zeta >= 0:
        zeta = min(zeta_max, zeta * 2)
    else:
        zeta /= 2
        if zeta > -zeta_min:
            zeta = zeta_min  # flip the sign
```

```python
# Bidirectional REPO-R (per token)
logp = new_logp.detach()  # latest model's logprobs + stop grad
if adv.item() > 0:
    adv = adv * (1 - zeta * logp)
    adv = adv.clamp_min(0.0)
elif adv.item() < 0:
    adv = adv * (1 + zeta * logp)
    adv = adv.clamp_max(0.0)
```

# E  QWEN 2.5 EXPERIMENTS

In previous publications, methods like LOOP performed well with "non-thinking" models such as Qwen 2.5 32B (Chen et al., 2025a). In our Qwen3 experiments however, LOOP (and very similarly, GRPO) experienced early entropy collapse and underperformed compared to other RL methods.

We conducted additional experiments (see Fig. 10 and Tab. 5) to determine whether this discrepancy arises from differences in model behavior or from implementation details. Key observations:

- Qwen 2.5 32B exhibits a significantly higher *initial* success rate (before the first training iteration) compared to Qwen 3 models. For example, on Test Normal the initial success rate is close to 40% versus under 10% for Qwen 3. We attribute this to Qwen3's tendency to produce excessively verbose thinking blocks which hinders the actual progress on the task.

- The best results on the hardest test split (Test Challenge) are substantially lower for Qwen 2.5 compared to Qwen 3, most likely reflecting the limitations of the respective base models.

- We were able to replicate and exceed results reported in previous work for Qwen 2.5 32B: the success rate of our best-performing LOOP checkpoints surpasses those in Chen et al. (2025a) by approximately 7% on Test Normal and 9% on Test Challenge. This improvement is most likely attributable to the numerical changes described in App. B, as our setup and hyperparameters for Qwen 2.5 closely match those in Chen et al. (2025a) in all other respects.

- Unlike in our Qwen 3 experiments, LOOP/GRPO do not experience rapid entropy collapse, whereas RLOO does, suggesting that base-model characteristics play a major role in entropy dynamics during training irrespective of the RL algorithm.

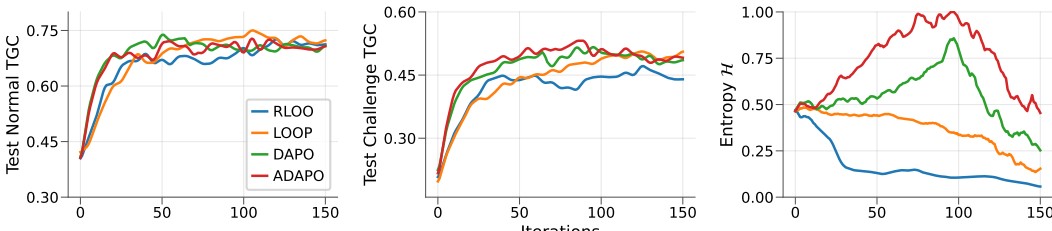

Figure 10: Qwen 2.5 32B test performance and token entropy on AppWorld vs. training iterations. Curves show mean across three independent seeds for each algorithm.

| Algorithm | Test Normal | Best TN | Test Challenge | Best TC |
|-----------|-------------|---------|----------------|---------|
| RLOO      | 0.72        | 0.78    | 0.47           | 0.50    |
| LOOP      | 0.75        | 0.78    | 0.50           | 0.54    |
| DAPO      | 0.74        | 0.77    | 0.50           | 0.56    |
| ADAPO     | 0.73        | 0.78    | 0.51           | 0.59    |

Table 5: Task-goal completion scores for AppWorld `Qwen-2.5-32B` by training algorithm. For each test split, we report the best average score across three seeds and the highest score among all seeds and training iterations.

