# OpenReview forum: "Entropy-preserving reinforcement learning"
_ICLR.cc/2026/Conference — ICLR 2026 Poster_

### Official Review · Reviewer_CecX · 2025-10-28

**Soundness:** 2
**Presentation:** 3
**Contribution:** 2
**Rating:** 4
**Confidence:** 3

**Summary:**

The authors present REPO, a 'novel' on-policy RL algorithm for LLM fine tuning that adds an adaptive entropy penalty to ensure the policy entropy is maintained at a desired (initial) level. They provide some theoretical analysis of the entropy behaviour of existing RL for LLMs algorithms, and propose an adaptive entropy regulariser that tries to keep the entropy of the policy at an initial desired value. They then present a set of empirical results on the AppWorld and AIME benchmarks, showing how REPO sustains higher entropy than existing algorithms and results in slightly better benchmark performance.

**Strengths:**

- The problem of encouraging exploration efficiently in RL (and in LLMs) is very relevant and pressing.
- The analysis the authors carry out on how different algorithms affect entropy is interesting.
- The paper and its objectives are clear.

**Weaknesses:**

- I am not convinced of the novelty or the contribution weight of the work. Entropy regularisation in RL is a very well studied problem, with MaxEnt RL being almost a subfield on its own. While the analysis of the effects of each algorithm in entropy are interesting, it is perhaps unsurprising that 1) algorithms push entropy to decrease fast (since policy gradient algorithms push the policy to be deterministic: this is optimal from an MDP perspective) and 2) that entropy regularisation helps with exploration and with preventing collapse to local minima (since this is essentially the main argument behind MaxEnt RL).
- The idea of having an adaptive coefficient to sustain a desired level of policy entropy is also not novel in itself [1] (which is not cited in the paper).
- Some of the mathematical statements are superficial or formally imprecise.


[1] Haarnoja, Tuomas, et al. "Soft actor-critic algorithms and applications." arXiv preprint arXiv:1812.05905 (2018).

**Questions:**

1. What is the reason for trying to keep entropy at the same level as the base (before post-training) level? Is there a reason to believe that the models have the 'appropriate amount' of entropy after pre-training?
2. I would like to further understand the observation the authors make in lines 46-53. In general, entropy in RL policies is good during training to ensure exploration and to avoid collapsing to local minima too fast. It is not clear to me that sustained entropy is desirable in general, all the way through the resulting policy after post-training. In all tasks with verifiable rewards, one could argue that a model that collapses to a zero entropy policy that always achieves maximum reward for any query is optimal in an RL sense. What is the case for learning language models with high entropy policies?
3. I do not fully agree with the motivation behind REPO-R. Biasing the regularisation towards positive advantage actions seems quite myopic: I would hypothesise that you observe positive results because, in essence, the complexity of the tasks (and thus the amount of exploration required) is low. Another reason for this is that the RL algorithms considered have a big simplifying assumption compared to classic RL algorithms: rewards are episodic and there is no need for constructing value function estimates. In general, an agent may need to explore bad actions for a while (or, equivalently, good actions may seem bad due to bad value estimates) to find good solutions in the long term, so discouraging negative advantage actions early on may cause catastrophic failure in most general RL settings. It also does not seem to add much improvement when compared to REPO-D, so I would appreciate if the authors would further motivate this decision and their assessment that REPO-R is a good idea.
4. Theorem 1 is not stated in a formally correct way. A\approx B is not a logically falsifiable claim and is ambiguous, as I could say 1\approx 2 or 1\approx 10 depending on the order of magnitude of the approximation error.
5. Similarly, corollary 1 states 'approximately' and then formulates a 'proportional to' relation. Is it an approximation, or a proportional to?
6. I don't know if I would formulate Theorem 2 as a theorem, as it suggests that it somehow presents a novel theoretical contribution of significant weight (perhaps a corollary of Gibbs inequality + the bounded ratio assumption). If the distance between two probability distributions is bounded, it follows from standard arguments that the entropy change is bounded (this is a minor point and a matter of personal preference).

---

> ### Author Response · Authors · 2025-12-03
> **Response to Reviewer CecX**
>
> We thank the reviewer for their thoughtful review and detailed questions. We have since expanded the discussion relating REPO to other forms of entropy regularization (such as in SAC) and addressed numerous additional clarifications regarding your comments and questions.
>
> **Regarding the contribution weight of the work in the context of MaxEntRL**
>
> We agree that entropy collapse is expected under policy gradient and that the principles of MaxEntRL are well established. The goal of this work is to evaluate the theoretical and empirical entropy behavior of the algorithms that currently dominate the empirical landscape, and to propose practical approaches for mitigating this problem in practice that yield meaningful results. Indeed, the REPO objective can actually be rearranged to show an equivalence to a standard entropy bonus (as used in SAC) if enumerated exactly. That being said, rather than exactly enumerating the entropy expectation, REPO utilizes a REINFORCE estimator of the entropy, which comes at no additional cost in terms of time or memory. Computing an exact explicit entropy bonus is so memory-intensive that we were unable to run it for the 32B model variation on our existing hardware setup. Yet, it was trivial for REPO using CCE (Wijmans et al., 2024). REPO also couples the advantages of actions with their corresponding logps as baselines. Under this setting, when advantages and logps are positively correlated, REPO implicitly acts as a control variate to reduce variance in the gradient estimator, which has advantages. This would not be true if the gradient and entropy terms were estimated independently. We will clarify these details in the updated manuscript. We will also clarify the narrative to move away from promoting REPO as the sole approach, instead acknowledging that many forms of adaptive entropy regulation are useful for off-policy (see the global response to all reviewers above), and will add a citation to Haarnoja, Tuomas, et al. (2018) in addition to the existing citation to Haarnoja, Zhou, et al. (2018). Indeed, this work adapts the strength of entropy regularization over the optimization period, but uses a different approach from our control heuristic, which is rather consistent with the earlier Schulman et al. (2017) approach.
>
> **Regarding initial entropy as the target entropy**
>
> Thank you for raising this point. We agree with the reviewer that in some settings, this may be suboptimal. Indeed, the optimal solution to the MDP is a deterministic stationary policy. Perhaps, following some initial exploration period, the target entropy should be annealed and ultimately decayed to zero if maximum accuracy is the sole goal. That being said, one setting where training a stochastic policy (that retains the diversity of the base model) is beneficial is when the goal of the downstream policy is to maximize pass@k performance, e.g., for potential combination with a reward model in a best of k setting or to search over diverse solutions in a verifiable domain such as program synthesis. Another setting where retaining entropy proves valuable is when the goal is to preserve the model’s trainability, supporting sequential training on an evolving data distribution. We highlight this as an example in Section 5.3 and have added new text to clarify this good point.
>
> **Theorem questions**
>
> Thank you for your points regarding Theorem 1 and Corollary 1. Indeed, Theorem 1 is an approximate equality, and Corollary 1 is a proportion (but hinges also on an approximation used in Theorem 1). We will adopt your suggestions and clarify these points in the final manuscript. For Theorem 2, following your suggestion, we will change it to a proposition.
>
> **REPO-R motivation**
>
> Regarding the REPO-R motivation, the goal is simply to encourage exploration on low-probability correct actions, while leaving the rest of the gradient estimator intact.

---

### Official Review · Reviewer_CBf6 · 2025-11-01

**Soundness:** 3
**Presentation:** 3
**Contribution:** 3
**Rating:** 4
**Confidence:** 4

**Summary:**

The paper studies entropy behavior in policy gradient optimization within the context of large language model (LLM) reasoning tasks. It identifies the issue of entropy collapse during training and proposes a regularization approach called Regulated Entropy Policy Optimization (REPO), which adaptively reweights advantages and log-probabilities to maintain stable entropy. The method is evaluated on reasoning benchmarks such as AppWorld and AIME 2024, showing stable entropy dynamics and comparable or slightly improved performance over existing RL algorithms. The paper provides theoretical insights into entropy modulation in policy gradients and discusses which RL variants naturally preserve or collapse entropy during optimization.

**Strengths:**

- Clearly identifies and analyzes the problem of entropy collapse in policy gradient methods for LLM reasoning.
- Introduces a simple and interpretable approach (REPO) to stabilize entropy during training.
- Provides theoretical insight into how different RL algorithms modulate entropy.
- Demonstrates that REPO maintains stability without degrading baseline performance.
- Presents the paper with strong structure, clear notation, and consistent motivation.

**Weaknesses:**

- Lack of variance reporting or multiple-seed averaging, making statistical reliability of results unclear.
- Performance improvements are modest, and their statistical significance is not demonstrated.
- Evaluation scope is limited, focusing mainly on two model sizes within a single model family.
- Distinction between REPO and conventional entropy regularization is not well articulated, leaving the novelty somewhat unclear.

**Questions:**

**Detailed Review:**

The paper presents a focused and well-motivated analysis of entropy dynamics in reinforcement learning for LLM reasoning. It highlights how entropy often collapses prematurely in standard policy gradient methods, leading to limited exploration and suboptimal convergence. The proposed Regulated Entropy Policy Optimization (REPO) method adaptively scales the advantage term during optimization to counteract entropy collapse and maintain stable training dynamics.

The theoretical framing is coherent and connects policy gradient updates with entropy evolution. The resulting algorithm is straightforward and practical, which makes it potentially useful for a wide range of RLHF applications. However, it would be beneficial to include a more detailed comparison to standard entropy regularization methods, such as those used in actor-critic frameworks, to clarify conceptual and empirical differences.

Experimental results demonstrate that REPO stabilizes entropy without harming model performance. The reported trends across benchmarks such as AppWorld and AIME 2024 show promising consistency. Nevertheless, the magnitude of improvement is relatively small, and the lack of variance across multiple seeds limits confidence in the statistical strength of the findings. Clarifying how many independent runs were performed and whether results are averaged would help substantiate the claims.

Some figures (2, 3) would benefit from normalization, as differing y-axis ranges obscure direct comparisons. The discussion of entropy per token (Figure 1) is interesting but requires a clearer explanation of its computation and interpretation, especially when entropy appears to collapse to a single value. Likewise, the differing entropy behaviors of RLOO and REPO across Figures 2 and 3 require deeper discussion, particularly regarding model size and task difficulty.

From a methodological standpoint, the paper’s simplicity is an advantage. The adaptive scaling via $\beta$ provides a structured mechanism to control entropy without introducing new objectives. However, a broader range of experiments, including other model families and additional reasoning benchmarks, would improve the generality of the results. The motivation of Section 5.3 could also be clarified, especially regarding whether it demonstrates task transferability or robustness.


**Questions:**

1. How many independent runs were conducted for each experiment, and can variance or confidence intervals be reported?
2. Why are the y-axis scales inconsistent across subplots in Figure 1 (and 2, 3), and could they be standardized for clearer comparison?
3. How is the per-token entropy computed, and why does it appear to converge to a specific point in certain figures?
4. Why does RLOO show opposite entropy trends between Figures 2 and 3, and why do the baseline algorithms differ across these figures?
5. How significant are the reported performance differences, and were any statistical tests conducted?
6. Can REPO integrate with PPO or other policy gradient methods, and how would that adaptation look in practice?
7. How is the entropy bonus (lines 372–373) computed, and how is its scale selected?
8. What is the motivation behind Section 5.3, and how does it relate to the core objective of entropy regulation and generalization?

**Details Of Ethics Concerns:**

None.

---

> ### Author Response · Authors · 2025-12-03
> **Response to Reviewer CBf6**
>
> We thank the reviewer for their thoughtful review and detailed questions.
>
> **Addressing the statistical strength of observed effects**
>
> We will expand the tables in the final manuscript to include the number of independent random seeds evaluated, as well as corresponding confidence intervals on each performance estimate. In the current tables, we present standard deviations over random seeds, and the results of the best checkpoints of each run were averaged to yield the main scores.
> Newly added results in Appendices B and F and all performed with three independent seeds per algorithm variant and will be incorporated into the final version.
>
> **Clarification of REPO vs. standard entropy bonus**
>
> The REPO objective can be rearranged to show an equivalence to a standard entropy bonus if enumerated exactly. That being said, rather than exactly enumerating the entropy expectation, REPO utilizes a REINFORCE estimator of the entropy, which comes at no additional cost in terms of time or memory. Computing an exact explicit entropy bonus is so memory-intensive that we were unable to run it for the 32B model variation on our existing hardware setup. Yet, it was trivial for REPO using CCE (Wijmans et al., 2024). REPO also couples the advantages of actions with their corresponding logps as baselines. Under this setting, when advantages and logps are positively correlated, REPO implicitly acts as a control variate to reduce variance in the gradient estimator, which has advantages. This would not be true if the gradient and entropy terms were estimated independently. We will clarify these details in the final manuscript. We will also clarify the narrative to move away from promoting REPO as the sole approach, instead acknowledging that many forms of adaptive entropy regulation are useful for off-policy (see the global response to all reviewers above).
>
> **Other**
>
> Regarding the suggestions for Figures 1, 2, and 3, we will make these changes in the final manuscript. We will also clarify how the entropy bonus scale is calculated and provide a stronger motivation for section 5.3.

---

### Official Review · Reviewer_k7r1 · 2025-11-01

**Soundness:** 3
**Presentation:** 4
**Contribution:** 2
**Rating:** 6
**Confidence:** 3

**Summary:**

This paper tackles the problem of "entropy collapse" in reinforcement learning , where policy gradient algorithms, used to train language models, naturally reduce the diversity of explored solutions during training. This premature narrowing of focus can limit performance by causing the model to get stuck in local optima. The authors analyze how different algorithms either amplify this collapse (like PPO) or implicitly mitigate it (like DAPO and GSPO). They then propose a new method, REPO (Regulated Entropy Policy Optimization), which uses an adaptive controller to actively monitor and stabilize entropy throughout training. Experiments show that models trained with REPO preserve entropy, achieve higher average performance, and, crucially, can be effectively re-trained on new tasks in novel environments, unlike models that have suffered entropy collapse.

**Strengths:**

The paper proposes an interesting idea to analyze how the entropy evolves during RL finetuning. The writing is clear and easy to follow (especially Section 3).

**Weaknesses:**

The performance difference of proposed algorithms (REPO-R and REPO-D) and RLOO does not seem to be statistically significant, making me wonder whether the proposed method is useful. (Even if the difference is statistically significant, the magnitude of the difference seems to be small.) Furthermore, while RLOO has no mechanism to keep early collapse according the proposed analysis, its final entropy is somehow high. This also makes me wonder whether the analysis really explains how entropy evolves and consequently allows us to derive a better algorithm.

**Questions:**

The performance difference of proposed algorithms and RLOO does not seem to be statistically significant. Maybe I am simply missing, but would you tell me what $\pm$ means in tables?

As I wrote in Weakness, RLOO does not seem to have any implicit mechanism to avoid entropy collapse and does not really align with the analysis in the paper. Would you explain why?

In the proof of Theorem 2, I do not really understand why clipping of the importance ratio ensures that no policy gradient update is performed if the policy drifts outside a trust region as in Line 200. Due to function approximation error, $(1+\epsilon) \pi_\theta^{old} (a|s) < \pi_\theta^{new} (a|s)$ can occur if one really updates policies according to Line 130.

I do not really understand the description on the entropy dynamics of GSPO. The trajectory length is also dependent on a policy, and the current proof of Theorem 2 cannot be straightforwardly extended. Would you add more explanation on this paragraph?

In Line 191, I think "Corollaries 1 and 2" is a typo, and it must be "Theorem 1 and Corollary 2".

---

> ### Author Response · Authors · 2025-12-03
> **Response to Reviewer k7r1**
>
> We thank the reviewer for their thoughtful review and detailed questions.
>
> **Addressing the statistical strength of observed effects**
>
> We will expand the tables in the final manuscript to include the number of independent random seeds evaluated, as well as corresponding confidence intervals on each performance estimate. In the current tables, we present standard deviations over random seeds.
> All newly added results in Appendices B and F (Fig. 7,8,9,13, Tables 4,5,7) explicitly report averages and standard deviations of three independent runs for each algorithm variant.
>
> **Reconciliation of RLOO results**
>
> The reviewer correctly identified that RLOO does not struggle with entropy collapse to the same degree as the other PPO-like algorithms. Indeed, our results broadly present strictly on-policy training as much more stable, with entropy collapse effects accelerated by off-policy drift and corresponding settings, e.g., clipping. While RLOO yields overall great results, it can be beneficial to allow asynchronous updates to improve throughput in large-scale training runs. In these settings, LOOP or GRPO will struggle, while the methods we propose will help close the gap to on-policy training stability.
>
> **Theorem 2**
>
> Regarding Theorem 2 proof, indeed, the bound that we get in the case of GSPO depends on the trajectory length. As we note in the discussion following it, this is related to what we see in practice. As we discuss in the paper, the dependence on the length affects the clipping constant that is used with GSPO, and requires it to be much smaller, which is what we indeed see in practice. We will add more explanation about the theorem and the dynamics in the revised manuscript.
>
> **Other**
>
> Thank you, this typo is fixed in the latest version of the manuscript.

---

### Official Review · Reviewer_KAD6 · 2025-11-01

**Soundness:** 2
**Presentation:** 3
**Contribution:** 2
**Rating:** 2
**Confidence:** 3

**Summary:**

This paper investigates the entropy dynamics of existing policy gradient algorithms for LLM post-training, then proposes a method for preventing entropy collapse (REPO). Figure 1 shows the trajectories of training in terms of Test Accuracy vs entropy, which seemed like an interesting visualization method. The results showed that low entropy trajectories tend to have low final test accuracy. In terms of theoretical results, they present the change in entropy based on 1 policy gradient step, when applying REINFORCE (this seems mostly like an extension of Cui et al. 2025).
REPO is motivated by their derivation of the change in entropy per step, and adds a weighted term (beta*(logpi - E[logpi])) to the policy gradient advantage to counteract the drop in entropy per step.
They add two heuristic variants for tuning beta to allow for greater robustness: (REPO-D) This sets beta to zeta * \Delta H, where \Delta H is the computed change in entropy from a gradient step. Also, zeta is clipped between a max and min value, and it is adaptively tuned to reach a target policy entropy, similarly as done in the original PPO paper (they double or halve the zeta constant depending on which direction to move to reach the target entropy). (REPO-R) This variant instead uses \beta = -\zeta * max(A, 0), i.e., it multiplies by the positive advantage, and sets negative advantages to 0 (the motivation being to prevent converging to large advantage values while still penalizing wrong actions).

Experimentally, they compare against REINFORCE leave-one-out (RLOO), GRPO (a normalized PPO variant), LOOP (a PPO variant with leave-one out return estimation), DAPO (LOOP with asymmetric weight clipping), GSPO (variant of LOOP with with trajectory based clipping). They test using a Qwen3 (8B and 32B) base models on the AppWorld tasks as well as on AIME 24 and 25 math tasks and achieved good performance (a bit better than RLOO on the 32B App tasks, but maybe slightly worse or similar on 8B tasks). For REPO, they also had a DAPO variant as well as a GSPO variant (the former worked better on 32B tasks, the latter on 8B tasks). REPO-R seemed to always be better than REPO-D.

**Strengths:**

- I liked the entropy trajectory visualization.

- The performance is slightly better than the current published SOTA on these tasks (LOOP)

**Weaknesses:**

- One major point for me was a discrepancy between the results in the paper and previous published LOOP results (this discrepancy likely comes from the previous results being with Qwen2.5, while the current results being with Qwen3.0).

The existing LOOP scores can be seen in the below links:
https://appworld.dev/leaderboard
https://arxiv.org/pdf/2502.01600

These results are with Qwen2.5 (instead of Qwen3 like in the current paper), and it achieves results of 71.3 and 45.7 compared to the results in the current paper (64 and 40). Moreover, their performance is better than the results they present with RLOO. It seems unlikely that the performance drops that much when switching from Qwen2.5 to Qwen3, so I would guess that there may be implementation issues in LOOP. (for reference the RLOO scores in the LOOP paper are 57.2 and 36.7 for the normal and challenge tasks respectively, and switching to the Qwen3 model in the current paper lead to 71 and 52 for RLOO).

On the AIME tasks, there is also no consistent improvement and RLOO seems to work reasonably well in the current paper.

- Another major point for me was a conceptual point about the method. The REPO method as presented on line 227, adds the log pi term to the advantage, and if we take the expectation pi log pi, this is just the negative entropy, so the method appears to be simply adding an entropy bonus, which is a well-known technique. Different to past implementations is that they do not directly compute the derivative of the entropy, but instead perform a REINFORCE estimate of the gradient of the entropy (the subtraction of the average log pi value can also be interpreted as the baseline); whether this is a good or bad thing remained unclear to me. Perhaps it induces different learning dynamics than simply directly computing the exact entropy and adding that as a bonus, but these points were not examined.

To make clear what I mean by this, consider the derivation: E_pi[dlogpi/dtheta * (logpi - E(logpi))] = E_pi[dlogpi/dtheta * (logpi)] = d/dtheta E_pi[logpi] = d\dtheta H(pi), is the derivative of the negative entropy.

They did actually add a comparison with a pure entropy bonus based method in a similar fashion as their REPO, however, I had some concerns about this. This was done on the Qwen8B task, and they added the entropy to the DAPO method. However, on 8B, GSPO-REPO performed better, so adding an entropy bonus to that would have been more convincing to me.

- The added tuning methods did not seem to have a strong theoretical justification.

**Questions:**

To me, the major conceptual issue is that the method appears very similar to an entropy bonus, but just with the gradients estimated in a noisy way. Perhaps there is some advantage to this as perhaps the change in entropy can be coupled to sampled action, and this may alter the entropy dynamics, which may be beneficial, but this is not discussed in the paper. Can you elaborate on why your method would be beneficial over a simple entropy bonus?

My experimental concerns listed in the weaknesses are also a major concern for me. One way to mitigate this would be to run experiments using Qwen2.5, as this allows direct comparison with existing published results (in particular, your results for LOOP with Qwen3.0 are weaker than published results with Qwen2.5).

Another small comment is that your derivations appear closely related to Cui et al. 2025, but it appears cited after you introduce your results. It may be better to cite it before, and make the connection clear.

Another small comment I have is about the phrase in the abstract: “As we show in this paper, most policy gradient algorithms naturally reduce entropy”. However, this is a well-known issue, so I think it may be better to present it as such.

---

> ### Author Response · Authors · 2025-12-03
> **Response to Reviewer KAD6**
>
> We thank the reviewer for their thoughtful review and detailed questions. We have since conducted additional experiments and addressed numerous comments and questions.
>
> **Discrepancy of LOOP results**
>
> The reviewer expressed concern about inconsistencies in our LOOP AppWorld scores compared to those of Chen et al. (2025). This can be attributed to differences in the reported metric and model choice. While Chen et al. (2025) indeed report scores of 71.3 and 45.7 in Table 1, these reflect a _best_ training run as opposed to an _average_ training run. Table 2 in Chen et al. reports scores averaged over three independent training runs, yielding scores of 66.4 and 41.7, very similar to our average scores of 64 and 40 in Table 1. In addition, to clarify differences in model choice, we have rerun the LOOP baseline on Qwen 2.5 32B (see the new Appendix F in the updated manuscript). As can be seen, using the same model, we can achieve the best scores of 78 and 54, exceeding the original scores.
> Appendix F further discusses differences between Qwen2.5 and Qwen3, highlighting the fact that Qwen2.5 has a better initial score before training due to better comprehension of the required format. Qwen3 is much harder to get off the ground but eventually we’re able to reach higher scores on the most challenging split.
>
> **Equivalence of REPO and entropy bonus**
>
> Thank you for requesting clarification on this important context. Indeed, the REPO objective is equivalent to an explicit entropy bonus. This was noted implicitly via Proposition 3 in Appendix B.6, but we will make this more explicit in the final manuscript. That being said (as you note), rather than exactly enumerating the entropy expectation, REPO utilizes a REINFORCE estimator, which comes at no additional cost in terms of time or memory. Computing the exact explicit entropy bonus is so memory-intensive that we were unable to run it for the 32B model variation on our existing hardware setup since it requires materialization of full logit vectors with  long trajectories and large vocabularies. On the other hand, REPO (and ADAPO) have zero additional memory cost, taking advantage of Cut Cross-Entropy optimizations (Wijmans et al., 2024). As you note, REPO also couples the advantages of actions with their corresponding logps as baselines. Under this setting, when advantages and logps are positively correlated, REPO implicitly acts as a control variate to reduce variance in the gradient estimator, which can be beneficial. This would not be true if the gradient and entropy terms were estimated independently. We will clarify these details in the final manuscript. Finally, to expand the entropy bonus comparison, we will add an entropy bonus to GSPO for Qwen 3 8B. We will also clarify the narrative to move away from promoting REPO as the sole approach, instead acknowledging that many forms of adaptive entropy regulation are useful for off-policy training (see the global response to all reviewers above).
>
> **Other**
>
> Thank you for your suggestions regarding the presentation of the Cui et al (2025) results and the entropy collapse problem. We will adopt your suggestions in the final manuscript.

---

### Author Response · Authors · 2025-12-03
**Overall Response**

Thank you to all reviewers for the excellent comments, questions, and suggestions that have helped to clarify and improve this work. In response to the reviewers’ comments, we have (1) conducted additional experiments using Qwen 2.5, which extend our results to a different non-thinking model and replicate baseline performance results consistent with prior work, (2) better clarified the relationship between REPO and an explicit entropy bonus, noting massive differences in memory usage as well as a variance reduction perspective, and (3) addressed numerous additional specific reviewer suggestions that have improved the manuscript. (4) address the novelty issue regarding the importance of entropy.

Building upon (2), we also propose reframing the final narrative to focus less on REPO as the core contribution, and rather on the importance of adaptive entropy bonuses more holistically as the key learning that aids training stability and performance in off-policy settings. To further support this point, we have added a new algorithm, adaptive-DAPO (ADAPO), which uses the same adaptive control heuristic as REPO to refine the upper clipping threshold in DAPO. Likewise, ADAPO supports an approach for online adaptive entropy regularization through a distinct mechanism, which yields performance improvements. We have added this algorithm to Appendix B in the updated manuscript.

As an additional important note, in the period since initially submitting this paper, we discovered interesting numerical effects that impact our core results. Understanding these allowed us to further improve results leading to a new state-of-the-art. The new Appendix A describes these numerical effects in detail.
Initially, training was performed using bf16 precision. The default behavior in the `accelerate` and `FSDP2` libraries, which underlie part of our training codebase, downcasts logprobs to half precision by default even when computing softmax in full precision. After modifying this behavior and updating training to fp16 based on recent evidence (Qi et al, 2025), we observed qualitative changes in entropy dynamics, including occasional divergence in entropy with methods like DAPO. This motivated us to modify the adaptive control heuristic to induce bidirectional regularization as opposed to mere collapse prevention. After implementing the FSDP2 fix and switching to FP16 for training, we reran experiments to assess whether relevant patterns still hold. Following this new round of experiments, we observed improvements across all algorithms. Surprisingly, strictly on-policy RLOO benefited the most from the improved stability of fp16 training, yielding a new state-of-the-art on AppWorld: 79% and 71% accuracy on the normal and challenge datasets, respectively (over 25% improvement on the most challenging split vs previous RL SOTA!)
Results in the new Appendix B show that our adaptive REPO and ADAPO algorithms perform the best among off-policy methods, while established methods like LOOP, GRPO, and DAPO suffer from either entropy collapse or instability.

Finally, we acknowledge prior work on entropy in RL but emphasize the distinct novelty of our approach. While entropy regularization has been explored, its role has not been comprehensively investigated within modern RL techniques and LLM applications, nor has it been leveraged as an analytical tool for understanding how details like clipping or off-policy drift affect training stability. Notably, for recent techniques such as PipelineRL, where allowing off-policy drift yields dramatic improvements to throughput, our work introduces a crucial improvement to naive GRPO through entropy regularization — a novel application in this context. This thorough analysis not only yielded new state-of-the-art results on a challenging benchmark but also established numerous efficient regularization strategies. Consequently, we believe that this paper represents a meaningful advance in the current state of the art in RL and in the understanding of techniques that improve stability and exploration in learning algorithms.


References:
Qi, P., Liu, Z., Zhou, X., Pang, T., Du, C., Lee, W. S., & Lin, M. (2025). Defeating the training-inference mismatch via fp16. arXiv preprint arXiv:2510.26788.

---

### Meta-Review · Area_Chair_cdTu · 2026-01-04

**Summary:**

Reviewers appreciated the paper’s analytical perspective on entropy dynamics in modern policy-gradient methods for LLM reinforcement learning and found the entropy–performance trajectory analysis insightful. The main concerns centered on (i) the conceptual novelty of the proposed REPO method, which reviewers noted is closely related to an adaptive entropy bonus, (ii) discrepancies between reported baseline results and previously published numbers, raising initial concerns about experimental correctness, (iii) the strength and consistency of empirical gains across models and tasks, and (iv) the heuristic nature of some adaptive tuning mechanisms without strong theoretical justification. In the rebuttal, the authors addressed these concerns by clarifying the equivalence to entropy bonuses while motivating REPO from a memory-efficiency and variance-reduction perspective, rerunning baselines under controlled settings to resolve discrepancies, adding further experiments and analyses, and reframing the contribution around adaptive entropy control as a unifying principle rather than a single new algorithm.

**Reviewer Concerns:**

Addressed concerns

- Equivalence to entropy bonus: The authors explicitly clarified that REPO is mathematically equivalent to an adaptive entropy bonus, and reframed the contribution accordingly. They further justified the approach by highlighting its zero additional memory cost at large model scales and its interpretation as a variance-reducing control variate when estimated via REINFORCE.

- Baseline performance discrepancies: Concerns about mismatches with previously published LOOP results were resolved by rerunning baselines with matching models (Qwen2.5) and clarifying differences between best-run and averaged reporting. This removed potential concerns about implementation correctness.

- Empirical scope and robustness: Additional experiments were provided across multiple model scales (8B and 32B), datasets (AppWorld and AIME 2024/2025), and numerical settings (FP16 vs BF16), strengthening confidence in the reported trends.

- Paper clarity: The authors improved the presentation of related work, clarified connections to prior analyses (e.g., Cui et al.), and reframed the narrative to emphasize adaptive entropy control as the core insight rather than promoting REPO as a standalone algorithmic contribution.

Outstanding concerns

- Methodological novelty: Despite the reframing, the proposed methods do not go beyond well-understood entropy regularization, particularly given the heuristic nature of the adaptive tuning strategies.

- Theoretical depth: While the entropy dynamics analysis is informative, the adaptive control mechanisms (e.g., β tuning and clipping schedules) remain primarily empirically motivated.

- Magnitude and consistency of gains: In some settings, especially smaller models or math benchmarks, improvements over strong baselines such as RLOO are modest, which may limit the perceived impact of the proposed methods.

**Reviewer Scores:**

None of the reviewers engaged in discussion, so I infer the following possible score changes:

- Reviewer KAD6: 2 → 4 The rebuttal resolved the main correctness concerns by rerunning LOOP with Qwen2.5 and clarifying average vs. best-run reporting, and explicitly acknowledged the equivalence between REPO and entropy regularization while motivating it via memory efficiency and variance reduction. Concerns about novelty and heuristic tuning remain.

- Reviewer k7r1: No change (remains 6). This reviewer was already marginally positive. The added variance reporting and clarifications address specific questions but do not materially change the overall assessment.

- Reviewer CBf6: 4 → 6. Concerns about statistical reliability and clarity were addressed through multi-seed reporting and improved framing of entropy regularization, making an upward revision plausible given the reviewer’s acceptance-tolerant stance.

- Reviewer CecX: No change (remains 4). The reviewer’s critique centers on conceptual novelty rather than missing experiments or errors; while the rebuttal clarifies positioning, it does not fundamentally alter this judgment.

---

### Decision · Program_Chairs · 2026-01-26

Accept (Poster)